# Discovery of L-threonine transaldolases for enhanced biosynthesis of beta-hydroxylated amino acids

Michaela A. Jones[1], Neil D. Butler[1], Shelby R. Anderson [1], Sean A. Wirt [1], Ishika Govil[1], Xinyi Lyu[2], Yinzhi Fang[2] & Aditya M. Kunjapur [1✉]

Beta-hydroxy non-standard amino acids (β-OH-nsAAs) have utility as small molecule drugs, precursors for beta-lactone antibiotics, and building blocks for polypeptides. While the L-threonine transaldolase (TTA), ObiH, is a promising enzyme for β-OH-nsAA biosynthesis, little is known about other natural TTA sequences. We ascertained the specificity of the TTA enzyme class more comprehensively by characterizing 12 candidate TTA gene products across a wide range (20-80%) of sequence identities. We found that addition of a solubility tag substantially enhanced the soluble protein expression level within this difficult-to-express enzyme family. Using an optimized coupled enzyme assay, we identified six TTAs, including one with less than 30% sequence identity to ObiH that exhibits broader substrate scope, two-fold higher L-Threonine (L-Thr) affinity, and five-fold faster initial reaction rates under conditions tested. We harnessed these TTAs for first-time bioproduction of β-OH-nsAAs with handles for bio-orthogonal conjugation from supplemented precursors during aerobic fermentation of engineered *Escherichia coli*, where we observed that higher affinity of the TTA for L-Thr increased titer. Overall, our work reveals an unexpectedly high level of sequence diversity and broad substrate specificity in an enzyme family whose members play key roles in the biosynthesis of therapeutic natural products that could benefit from chemical diversification.

[1] Department of Chemical and Biomolecular Engineering, University of Delaware, Newark, DE 19716, USA. [2] Department of Chemistry and Biochemistry, University of Delaware, Newark, DE 19716, USA. ✉email: kunjapur@udel.edu

Aryl non-standard amino acids (nsAAs) that contain a hydroxyl-group on the β-carbon are found naturally in many highly effective antimicrobial non-ribosomal peptides (NRPs) such as vancomycin[1], ribosomally synthesized and post-translationally modified peptides (RiPPs) such as ustiloxin B[2], and industrially as small molecule antibiotics and therapeutics such as amphenicols[3,4] and droxidopa[5]. Beyond their current natural and industrial uses, aryl beta-hydroxy non-standard amino acids (β-OH-nsAAs) share structural similarity with nsAAs used for genetic code expansion[6], a technology that has had a profound impact on chemical biology and drug development. Efficient enzymatic synthesis of stereospecific β-OH-nsAAs could pave the way for inexpensive, one-pot production of chemically diverse ribosomal and non-ribosomal peptide products (Fig. 1a). Chemical diversification is valuable for drug development for purposes such as improving cell permeability[7], maintaining effectiveness[8], and increasing potency[9]. Further, fermentative, one-pot production of β-OH-nsAAs could enable their integration into more complex products like NRPs, RiPPs, and proteins, which are typically produced through fermentation because of their high requirements for protein synthesis and co-factor regeneration[10]. Until recently, strategies for the biosynthesis of β-OH-nsAAs in cells were limited by restricted substrate specificity or thermodynamic favorability. Many naturally occurring β-OH-nsAAs are produced within NRP synthase complexes in which the active enzyme performing the beta-hydroxylation is highly specific, or post-translationally in RiPPs by hydroxylases which are poorly characterized enzymes, limiting the potential for product diversification[11–13]. Alternatively, threonine aldolases (TAs) are a well-established enzyme class that exhibits substrate promiscuity and have been engineered to maintain high stereospecificity for β-OH-nsAAs production[14–16]. However, TAs naturally favor the decomposition of β-OH-nsAAs and require high concentrations of glycine for efficient product formation, limiting their use in fermentation.

A novel enzyme class known as L-threonine transaldolases (TTAs) can perform similar chemistry to TAs with low reversibility, high stereoselectivity, and high yields. TTAs are fold-type I pyridoxal 5'-phosphate (PLP)-dependent enzymes that catalyze the retroaldol cleavage of L-threonine (L-Thr) to form acetaldehyde and a glycyl-quinonoid intermediate that then reacts with an aldehyde acceptor to form a β-OH-nsAA. Interestingly, TTAs have higher sequence similarity to serine hydroxymethyltransferases (SHMTs) which naturally catalyze the formation of serine from glycine[17]. Three types of TTAs have been identified: fluorothreonine transaldolases (FTases)[18] that act on fluoroacetaldehyde acceptors; threonine:uridine 5' aldehyde transaldolases (LipK, AmbH)[19,20] that act on uridine 5' aldehyde acceptors; and L-TTAs that act on aryl aldehyde acceptors. In 2017, the TTA known as ObiH (or ObaG) was discovered as a part of the obafluorin biosynthesis pathway that natively catalyzed the aldol-like condensation of L-Thr and 4-nitrophenylacetaldehyde to produce the corresponding β-OH-nsAA (Fig. 1b)[21,22]. Since its discovery, ObiH and a 99% similar variant, PsLTTA, have been characterized to exhibit activity on over 30 aldehyde substrates as a purified enzyme and in resting cell biocatalysts, with notably little to no activity on aromatic aldehydes that contain strongly electron-donating functional

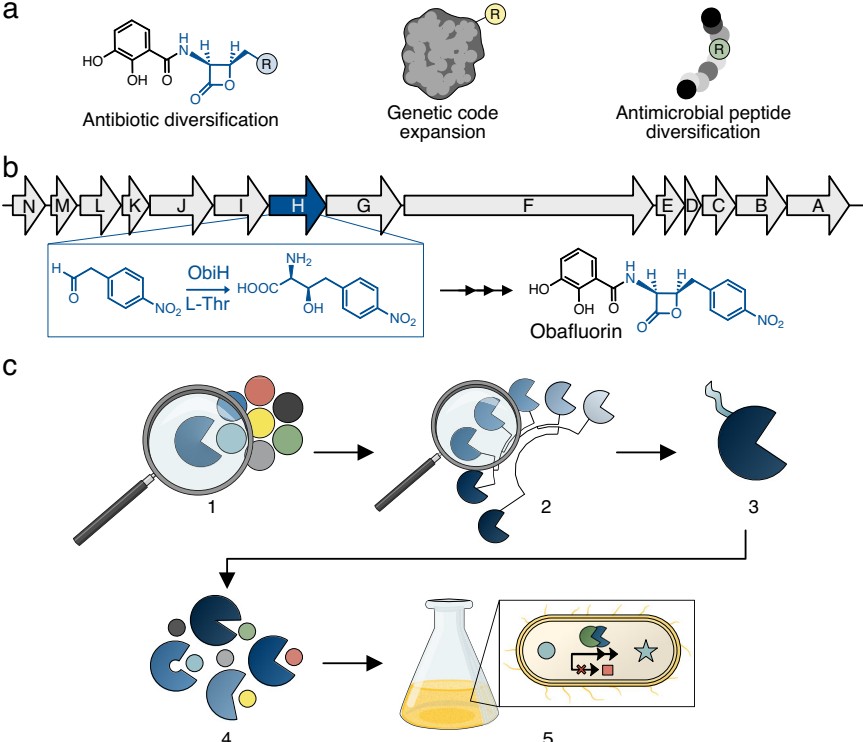

**Fig. 1 Threonine transaldolases are promising enzymes for the biosynthesis of chemically diverse β-OH-nsAA products. a** Depiction of potential applications for β-OH-nsAAs including diversified antibiotics, genetic code expansion, and novel non-ribosomal peptides. **b** Depiction of the natural biosynthetic gene cluster from *Pseudomonas fluorescens* that is responsible for the biosynthesis of the antibiotic obafluorin. One of the key enzymes in this pathway is ObiH, a threonine transaldolase (TTA) discovered in 2017. **c** Schematic of what we investigated in this study: (1) ObiH activity on multiple candidate substrates; (2) Bioprospecting for candidate TTAs of lower protein sequence identity than previous efforts; (3) A genetic strategy to improve TTA expression; (4) The biochemical characterization of candidate TTAs in regard to substrate scope and L-Thr affinity; (5) The potential for TTA-catalyzed formation of beta-hydroxylated non-standard amino acids during aerobic fermentation using an engineered chassis for aldehyde stabilization.

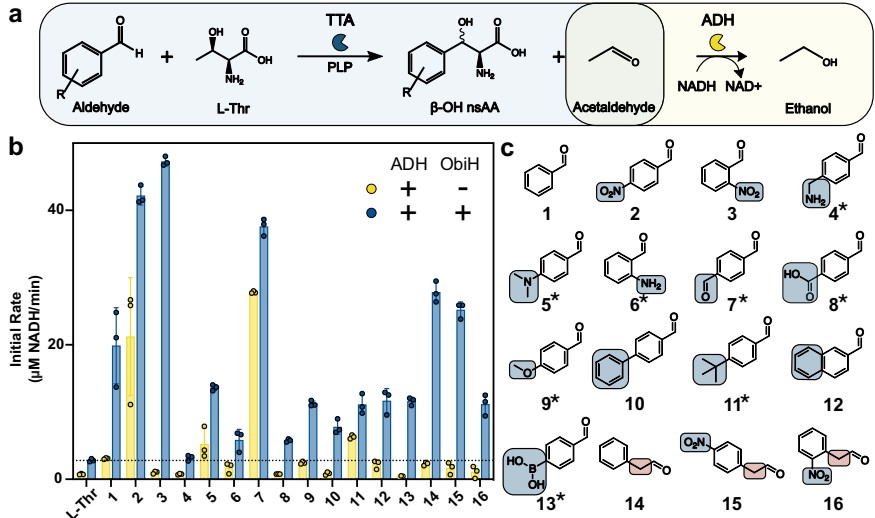

**Fig. 2 Use of a TTA-ADH coupled assay for screening activity of ObiH on a diverse array of aromatic aldehyde substrates. a** Reaction schematic for coupled enzyme reaction that enables reaction monitoring at 340 nm if appropriate conditions and controls are used. Important negative controls are no addition of aldehyde (to account for the rate of L-Thr decomposition) and no addition of ObiH (to account for potential ADH-catalyzed reduction of the aldehyde substrate). **b** Initial rates of ObiH on aldehyde substrates relative to an L-Thr background measurement and ADH background activity on aldehydes. The horizontal line indicates the L-Thr background decomposition observed in the TTA-ADH coupled assay. Any activity greater than the dotted line and the corresponding ADH activity is considered a successful activity of the TTA on that aldehyde. The experiment was performed in triplicate with each replicate displayed as an individual data point and error bars representing standard deviations. **c** Chemical structures of the aldehydes investigated in this study. Asterisks indicate substrates never previously screened with TTAs to our knowledge.

groups[23–28]. In these contexts, ObiH was shown to maintain low reversibility and high stereospecificity with a preference for the *threo* diastereomer, the isomer found in many natural products[1,29,30]. ObiH, and TTAs more broadly, are promising alternatives to produce chemically diverse β-OH-nsAAs. While ObiH expresses well in heterologous hosts like *Escherichia coli*, it has reported limitations in substrate scope, has a low L-Thr affinity, and has not been studied in fermentative conditions. Further, the aldehyde substrates for ObiH are unstable and potentially toxic in live cell contexts.

To address these challenges, we sought to further characterize ObiH, the products of other naturally occurring genes whose translations share similarity with known TTAs, and the ability of TTAs to form β-OH-nsAAs during heterologous expression in cells grown under aerobic conditions. At the outset of our study, ObiH, PsLTTA (a 99% similar homolog)[26], and a promiscuous FTase (FTaseMA)[31], were the only TTAs characterized to act on aromatic aldehydes. Furthermore, early studies[26,27] did not report testing of certain potentially useful aldehydes such as those that contain large hydrophobic moieties for cell penetration[7] or handles for bio-orthogonal click chemistry[32–34]. Additionally, the reported $K_M$ of ObiH for L-Thr ($40.2 \pm 3.8$ mM[22]) would suggest that the reaction would not proceed well in fermentative conditions without supplementation of L-Thr since natural *E. coli* L-Thr concentrations are low (normally <200 μM[35]). Interestingly, LipK and FTaseMA were reported to have lower L-Thr $K_M$ (29.5 mM[19] and 1.2 mM[31], respectively), but both are reported to have poor soluble expression in *E. coli*. Together, these observations offer promise for identifying a natural TTA that accepts a broad aldehyde substrate scope, has a high L-Thr affinity, and is active in heterologous host *E. coli*. Very few TTAs have been identified in nature, and many are likely annotated as hypothetical proteins or SHMTs based on their primary amino acid sequence.

In this paper, we addressed each of the challenges associated with engineering in vivo biosynthesis of β-OH-nsAAs in a model heterologous host: low L-Thr affinity, protein solubility in *E. coli*, and aldehyde substrate stability (Fig. 1c). To enable rapid

screening of many aldehydes and enzymes, we first optimized a high-throughput in vitro assay for characterization of TTAs on diverse aldehydes and demonstrated activity of ObiH on aldehydes that contain handles for bio-orthogonal conjugation. To explore the natural TTA sequence space, we then generated a sequence similarity network (SSN) of enzymes with high similarity to ObiH, FTase, and LipK. After appending a solubility tag to many distantly related TTAs, we observed dramatically improved enzyme expression and identified previously unreported TTAs that exhibit higher L-Thr affinity, faster reaction kinetics, and broad substrate scope. Remarkably, one of the best TTAs tested is annotated as a hypothetical protein and shares only 27.2% sequence identity with ObiH. Next, we biosynthesized β-OH-nsAAs by expressing the TTAs in cells that were engineered for aldehyde stabilization, and we coupled the TTAs to a carboxylic acid reductase (CAR) to limit toxic aldehyde accumulation. Finally, we demonstrated the activity of several CARs and a TTA in vitro and in growing cells to produce 4-azido-β-OH-phenylalanine (4-azido-β-OH-Phe), an nsAA with a well-established handle for bio-orthogonal conjugation. Our work brings the field closer to achieving a one-pot synthesis of chemically diverse peptides and proteins through the biosynthesis of β-OH-nsAAs in cells growing in aerobic conditions after supplementation with aldehyde or acid precursors.

## Results

**Optimizing a high-throughput assay for screening TTA activity on diverse aldehydes.** To expand our understanding of the TTA enzyme class, we wanted a high-throughput method for rapid screening of multiple enzymes and candidate aldehyde substrates. We began by analyzing a previously reported coupled enzyme assay (Fig. 2a and S1) based on the addition of an alcohol dehydrogenase (ADH), which consumes NADH to reduce the co-product acetaldehyde in a manner that can be monitored at 340 nm[19,25,36]. Unfortunately, this coupled assay for TTA activity suffers from false positives and confounding variables which we sought to address. First, the commercially available ADH from

*Saccharomyces cerevisiae* (ScADH) exhibits activity on many aromatic aldehydes which were candidate substrates for ObiH (Fig. S2a). We briefly investigated other ADHs from *E. coli* to attempt to identify an alternative that might limit this undesired activity while remaining active on the desired acetaldehyde co-product, but we did not identify a better ADH (Fig. S3). To address the false positives observed from ADH activity on the aldehyde acceptor, we optimized the concentrations of the ADH and aldehyde used in the reaction, and we introduced a control in which only the ADH and substrate were present ("no TTA"). Second, the characterized TTAs are known to catalyze the decomposition of L-Thr in the absence of an aldehyde substrate, which is an undesired reaction that also generates an acetaldehyde co-product and thus another false positive[37] (Fig. S2b). To account for the background production of acetaldehyde by the TTA with L-Thr, we introduced a control in which the reaction contained TTA and ADH but lacked an aldehyde substrate ("L-Thr"). Another limitation of the TTA-ADH coupled assay is that many of the aromatic aldehyde candidate substrates absorb at the same measurement wavelength which we accounted for by using low aldehyde concentrations (Table S1). With each limitation addressed, we validated the TTA-ADH coupled assay by performing high-performance liquid chromatography (HPLC) analysis, using the chemically synthesized β-OH-nsAA standard for the assumed product from **3**, over a time course where we observed that the addition of the ScADH improves reaction rates three-fold (Figs. S4–6). As previously reported by others[25,36], we were also able to improve β-OH-nsAAs yields when using the ScADH coupled to a co-factor regeneration system (Fig. S7). As the last step of verification, we screened the TTA-ADH coupled assay with ObiH before and after photo-treatment[37]. We observed no differences in reaction rate between the two photo-treatment conditions and continued to assay the TTAs without photo-treatment (Fig. S8).

Upon assay validation, we sought to rapidly probe the activity of ObiH on diverse aldehydes to expand the potential chemical handles of β-OH-nsAAs. We successfully screened ObiH against 16 unique substrates in a single experiment (Fig. 2b, c). We validated the activity of ObiH on substrates like the native substrate, 4-nitro-phenylacetaldehyde (**15**), and 2-nitro-benzaldehyde (**3**), which ObiH has been reported to exhibit high activity on. Our screen included nine substrates not previously tested with ObiH to our knowledge; activity on seven of these substrates was confirmed with new peak formation via HPLC or LC-MS (Figs. S9–S20). These substrates include aldehydes that contain amines, conjugatable handles, or larger hydrophobic groups to improve the chemical diversification of β-OH-nsAA products. Our results supported the known general trend[23,26] that aldehydes containing electron-withdrawing ring substituents are the preferred substrates of ObiH. As expected, the amine-aldehydes were very poor substrates for ObiH, which we hypothesize is because of the strong electron-donating potential of amines. Despite the observed trend that ObiH does not accept aldehydes containing strongly electron-donating ring substituents, we did observe that there was some activity on aldehydes with moderate electron-donating potential like 4-methoxy-benzaldehyde (**9**), 4-biphenylcarboxaldehyde (**10**), and 2-napthalaldehyde (**12**). Activity on larger, hydrophobic substrates is promising because these substrates can be used to modulate cell permeability for peptides. Additionally, we observed the activity of ObiH on terephthalaldehyde (**7**) and 4-boronobenzaldehyde (**13**) which both contain groups that can serve as bioconjugatable handles. With these results, we hypothesized that the TTA-ADH coupled assay can provide a broad and deep initial lens into the functional characterization of this under-explored enzyme class when used under appropriate conditions and with important controls.

**Bioprospecting for diverse, putative TTAs**. We used bioprospecting as an approach to advance our understanding of the TTA enzyme class and potentially discover a TTA capable of overcoming the limitations of ObiH such as its low affinity for L-Thr. Using a protein sequence similarity network (SSN) that was generated with over 800 sequences produced from a BLASTp search of ObiH, LipK, and FTase, we selected 12 additional putative TTAs (Fig. 3a). We selected five putative TTAs from the same cluster as ObiH, all exhibiting >50% sequence identity to ObiH, in addition to seven randomly selected putative TTAs from clusters with 20%-30% sequence identity to ObiH[38] (Fig. 3b and S21). For one enzyme from the ObiH cluster, we arbitrarily cloned a variant to contain a 36-residue truncation from the N-terminus (StTTA-Δ36) such that its new N-terminal residue would align with the sequence of ObiH and the other candidate TTAs. RaTTA and SNTTA were selected from the cluster containing LipK, DbTTA from the cluster containing FTase, and TmTTA from the cluster containing sequences annotated as SHMTs. Lastly, three TTAs (NoTTA, PbTTA, and KaTTA) were selected from distinct clusters with no characterized enzymes. The broad range of sequence identity of candidate TTAs from 20 to 80% with respect to ObiH and to each other indicates a broader sampling of the TTA-like sequence space in any one study than past efforts to our knowledge.

Upon selecting our list of candidate TTAs, we proceeded to test the heterologous expression of codon-optimized genes in *E. coli* for purification and in vitro biochemical characterization. Given the reported difficulty of expressing LipK and FTases[19,31], we were not surprised to observe little to no expression of the TTAs from the clusters containing FTase and LipK; however, we also observed low expression of TTAs from unexplored clusters, and unexpectedly, two from the cluster containing ObiH. Simple non-genetic methods for improving protein expression like changing culture temperature were unsuccessful. Instead, we hypothesized that the appendage of a small solubility tag, the Small Ubiquitin-like Modifier motif (SUMO tag)[39,40], could improve expression. We observed that the tag dramatically improved the expression of 11 TTAs (Fig. 3c and S22). To create the option of removing the SUMO tag if it were to impact activity, we cloned a TEV protease site[41] between the SUMO tag and each TTA gene. With the addition of the SUMO-tag, we successfully purified nine TTAs for further screening. Interestingly, we only observed the vibrant pink color characteristic of ObiH[22,37] with PiTTA, BuTTA, and s-KaTTA. All other TTAs had a very faint pink color or no coloration at all under the expression conditions we tested.

**Screening and characterization of bioprospected TTAs**. Once we purified the putative TTAs, we screened them for aldol-like condensation activity. We first screened each purified enzyme with the SUMO tag fusion intact using the TTA-ADH coupled assay. Our choice to characterize SUMO-tagged proteins was well justified for three reasons: (1) the predicted structures generated with AlphaFold2[42] suggested the N-terminal region is distal from the active site for all TTAs screened; (2) the ultimate goal was to identify better homologs for expression under fermentative conditions where tag removal would be too complex or resource intensive; (3) we tested one TTA with and without the SUMO tag to verify that the tag did not impact activity (Fig. S23). We then screened each purified enzyme using the TTA-ADH coupled assay with 2-nitro-benzaldehyde, **3**, the best-performing substrate from the screen of ObiH that was not a substrate of the ScADH. We observed that five enzymes (PiTTA, CsTTA, BuTTA, s-KaTTA, and PbTTA), had activity comparable to or better than ObiH (Fig. 4a).

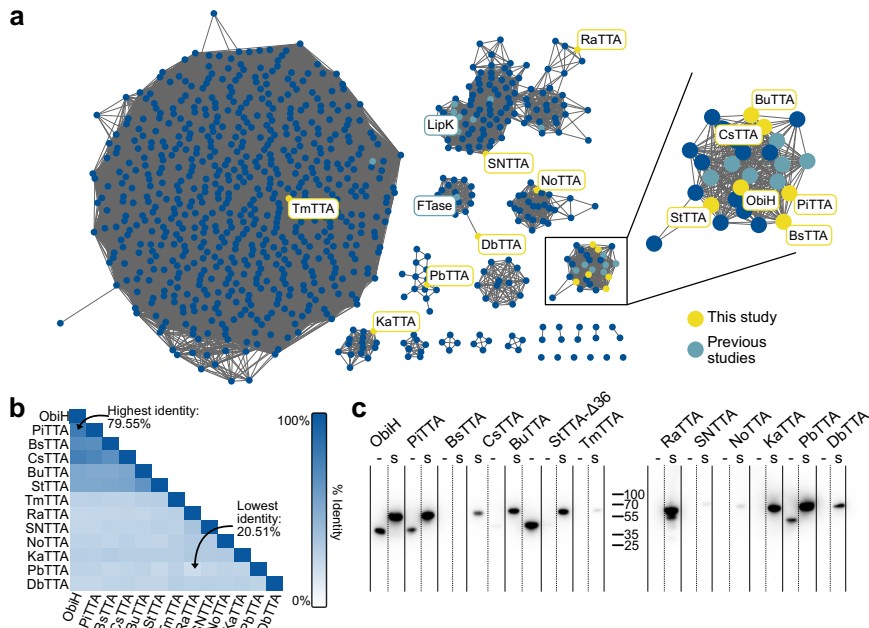

**Fig. 3 Bioprospecting and expression of putative threonine transaldolases. a** A Protein Sequence Similarity Network (SSN) containing 859 sequences related to ObiH, LipK, and FTase with selected putative TTAs highlighted in yellow. Existing enzymes characterized in the literature are highlighted in teal except those found in the largest cluster which contains many SHMTs. **b** Sequence identity matrix for all selected TTAs in this study. **c** Western blot of all TTAs with the tagged and untagged TTA constructs demonstrating improved expression of TTAs with a SUMO solubility tag. Proteins that contain an N-terminal SUMO tag followed by a TEV protease cleavage site, and no other changes, are shown in lanes indicated by the 's'.

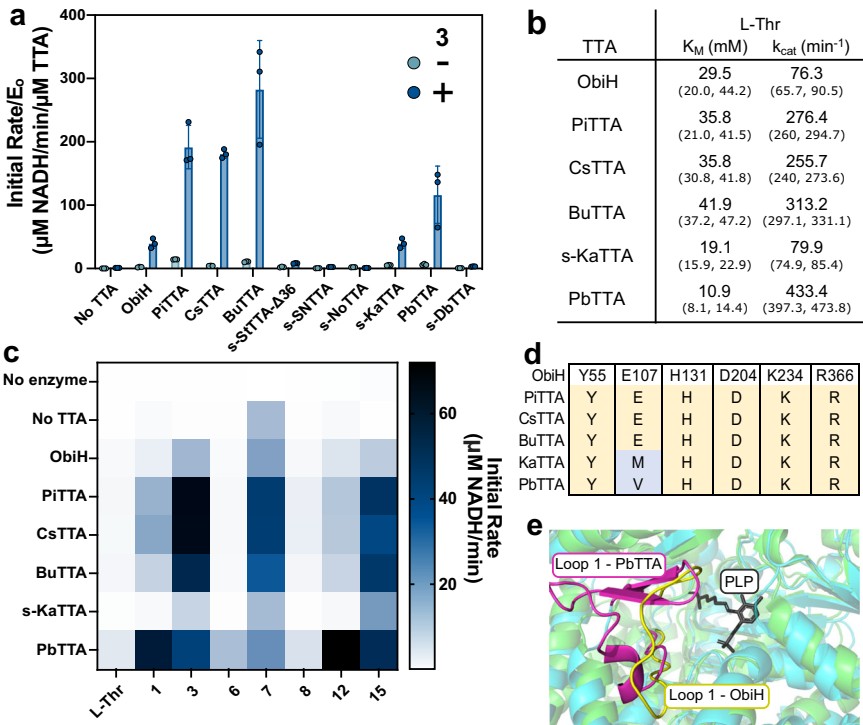

**b**

| TTA | L-Thr $K_M$ (mM) | L-Thr $k_{cat}$ (min$^{-1}$) |
|---|---|---|
| ObiH | 29.5 (20.0, 44.2) | 76.3 (65.7, 90.5) |
| PiTTA | 35.8 (21.0, 41.5) | 276.4 (260, 294.7) |
| CsTTA | 35.8 (30.8, 41.8) | 255.7 (240, 273.6) |
| BuTTA | 41.9 (37.2, 47.2) | 313.2 (297.1, 331.1) |
| s-KaTTA | 19.1 (15.9, 22.9) | 79.9 (74.9, 85.4) |
| PbTTA | 10.9 (8.1, 14.4) | 433.4 (397.3, 473.8) |

**d**

| ObiH | Y55 | E107 | H131 | D204 | K234 | R366 |
|---|---|---|---|---|---|---|
| PiTTA | Y | E | H | D | K | R |
| CsTTA | Y | E | H | D | K | R |
| BuTTA | Y | E | H | D | K | R |
| KaTTA | Y | M | H | D | K | R |
| PbTTA | Y | V | H | D | K | R |

**Fig. 4 Characterization of putative threonine transaldolases. a** Screen of all purified TTAs using the TTA-ADH coupled assay on **3**. The experiment was performed in triplicate with each replicate as an individual point. Error bars represent standard deviations. **b** Apparent L-Thr $K_M$ and $k_{cat}$ measurements for TTAs that exhibited activity greater than or equal to ObiH calculated using non-linear regression. Parenthetical values represent the 95% confidence interval. **c** Heatmap showing initial rates for six active TTAs against multiple aromatic aldehyde substrates. **d** Multi-sequence alignment of the predicted conserved catalytic residues for the six active TTAs. **e** Superimposed structure and predicted structure illustrating the Tyr55-Pro71 loop region of ObiH compared to the predicted equivalent region for PbTTA. ObiH is represented in green with the loop region highlighted in yellow with the PLP highlighted in gray indicating the region of the active site. The PbTTA is indicated in blue with the corresponding loop region highlighted in pink.

Given the activity of these distantly related enzymes and their annotation as SHMTs or hypothetical proteins, we wanted to further validate the amino acid substrate specificity of the active enzymes and further screen the inactive TTAs. We performed an in vitro assay over 20 h using **3** as the aldehyde substrate and either L-Thr, Glycine (Gly), or L-Serine (L-Ser) as the candidate amino acid. Since the TTA-ADH coupled assay is specific to L-Thr, we analyzed TTA activity via HPLC with a chemically synthesized β-OH-nsAA standard for the assumed product from **3**. We confirmed that the active purified TTAs (PiTTA, CsTTA, BuTTA, s-KaTTA, and PbTTA) only act with L-Thr with no β-OH-nsAA formation using L-Ser or Gly (Fig. S24). Further, this result confirmed that after 20 h, ObiH, PiTTA, CsTTA, BuTTA, s-KaTTA, and PbTTA all approached 100% conversion of the aldehyde to the final β-OH-nsAA product. s-KaTTA and PbTTA produce almost stereochemically pure isomers of the *threo* β-OH-nsAA with de value of 97% and 98%, respectively, which is better than the de value of 80% for products from ObiH (Fig. S25). Of the inactive enzymes (NoTTA, TmTTA, DbTTA, and StTTA-Δ36), we observed that StTTA-Δ36 was active with the formation of the β-OH-nsAA product from **3** and L-Thr, suggesting it is too slow to detect using the TTA-ADH coupled assay. NoTTA, TmTTA, and DbTTA yielded no product, which leaves the possibility that they could be TTAs that do not accept **3** under the conditions tested or that they may not be TTAs.

To explore the possibility that DbTTA and TmTTA are TTAs active on other related aldehydes, we sought to examine their activity with L-Thr and aldehyde substrates with different ring substituent positions (**2**), bulkier, hydrophobic chemistry (**10**), and aldehyde chain length (**14**) using the TTA-ADH coupled assay. Neither of these proteins appeared to have any TTA activity, nor the reported L-Thr decomposition activity (Fig. S26). We did not perform this analysis for NoTTA because we did not observe L-Thr decomposition activity, and this was predictive of inactivity on the additional substrates for both DbTTA and TmTTA.

For those enzymes with comparable or faster activity than ObiH, we next sought to determine their affinity for L-Thr, which we obtained by performing the TTA-ADH coupled assay at different L-Thr concentrations and a non-saturating phenylacetaldehyde concentration of 1 mM (Fig. 4b, and S27). Notably, our assay yielded a lower $K_M$ for ObiH towards L-Thr, 29.5 mM, than the literature value (40.2 ± 3.8 mM). Two differences between our assays were the substrate, phenylacetaldehyde (**14**) instead of 4-nitrophenylacetylaldehyde (**15**), and the assay format, ADH coupling rather than a discontinuous HPLC assay. We used phenylacetaldehyde for the enzyme kinetics assay because it does not interfere with the absorbance at 340 nm, is structurally similar to the previously reported substrates for TTA screening, and is a low enough concentration to avoid observing background ADH activity. While we choose phenylacetaldehyde for this investigation, we believe this analysis could be performed with many different aldehyde substrates and may yield distinct kinetic parameters. Because a live cellular environment would also contain alcohol dehydrogenases for the reduction of acetaldehyde, it is possible that the $K_M$ values that we are measuring using the TTA-ADH coupled assay may be more realistic for our envisioned applications. Encouragingly, under these conditions, we observed that s-KaTTA and PbTTA have lower L-Thr $K_M$ than ObiH (19.1 mM and 10.9 mM, respectively). Interestingly, many of our TTAs such as PiTTA, CsTTA, BuTTA, and PbTTA have higher measured L-Thr $k_{cat}$ values than ObiH using phenylacetaldehyde as the aldehyde substrate (Fig. 4b). Thus, each of the characterized enzymes is faster or has higher affinity for L-Thr than ObiH does and may prove to be improved alternatives to ObiH depending on the desired application.

Given the broad substrate scope of ObiH, we sought to examine a set of aryl substrates that would span the spectrum of electronic properties and include some that ObiH exhibits little to no activity on. By providing a set of seven substrates to all six TTAs, we aspired to help elucidate the landscape of specificity within this family while possibly identifying variants that exhibited higher activity or altered specificity (Fig. 4c). We specifically selected substrates with ring substituents with different electron-withdrawing properties (**1**, **3**, **6**, **7**, **8**), substituent size (**12**), and aldehyde chain length (**15**) to compare the activity of the putative TTAs to ObiH. We observed several interesting activities—for example, the TTAs that appeared to have higher $k_{cat}$ values in the ObiH cluster, such as PiTTA and BuTTA, remain relatively selective and are both reported to be a part of biosynthetic gene clusters for obafluorin[43] (Table S2). Additionally, one of the most active TTAs, PbTTA, also maintains high activity on a diverse array of substrates, originates from a different cluster of the SSN as ObiH, and exhibits low sequence identity (27.2% identity). This suggests that the TTA enzyme family may be broader than previously thought, with many more active homologs worthy of characterization for the elucidation of natural products or for applications in biocatalysis and synthetic biology.

**Comparative sequence analysis for characterized TTAs**. To help shed some light on the potential molecular basis for substrate specificity, we performed a comparative sequence analysis of the active TTAs with a focus on known residues implicated in catalysis (H131, D204, K234) or PLP-stabilization (Y55, E107, and R366) in ObiH, as well as two loop regions that are reported to contribute to substrate specificity[37]. We performed a multiple sequence alignment across the enzymes selected and a series of characterized fold-type I PLP-dependent enzymes, including LipK from *Streptomyces sp.* SANK 60405[19], FTase from *Streptomyces cattleya*[18], and SHMT from *Methanocaldococcus jannaschii*[44] (Fig. S28). Many of the active TTAs within the ObiH cluster had the same residues at these sites. However, PbTTA and KaTTA appeared to have a modified residue at E107 which is reported to perform hydrogen bonding for PLP stabilization (Fig. 4d). This was not surprising as this residue is not conserved across related PLP-dependent enzymes. Further, we evaluated two loop regions from ObiH between Tyr55 and Pro71 (loop 1) as well as Glu355 and His363 (loop 2) that are reported to contribute to substrate specificity given their role in SHMTs as folate binding regions[45]. While loop 1 appears to be composed of different residues across the TTAs screened, PbTTA has a unique 11 amino acid insertion in the equivalent loop 1. We then aligned the published ObiH crystal structure with an AlphaFold prediction for PbTTA and observed a β-sheet within loop 1 of PbTTA (Fig. 4e). In contrast, loop 1 in ObiH is relatively unstructured and published MD simulations[37] of ObiH suggest loop 1 is highly flexible.

Since this enzyme class is recently discovered, we wanted to explore the unique sequence properties of each cluster to determine if there are any distinguishing features across clusters. By examining each cluster one at a time and aligning all sequences within each cluster to ObiH, we identified that catalytic residues (H131, D204, and K234) are conserved across the clusters containing ObiH, LipK, FTase, KaTTA, and PbTTA (Fig. S29). Further, R366 is highly conserved (>90%) for all clusters analyzed. As highlighted for KaTTA and PbTTA, E107 is not conserved. For E107, each cluster appeared to have a different predominant residue in that position. Additionally, given the distinction between loop 1 of ObiH relative to SHMTs and PbTTA, we wanted to explore the sequence context of this loop

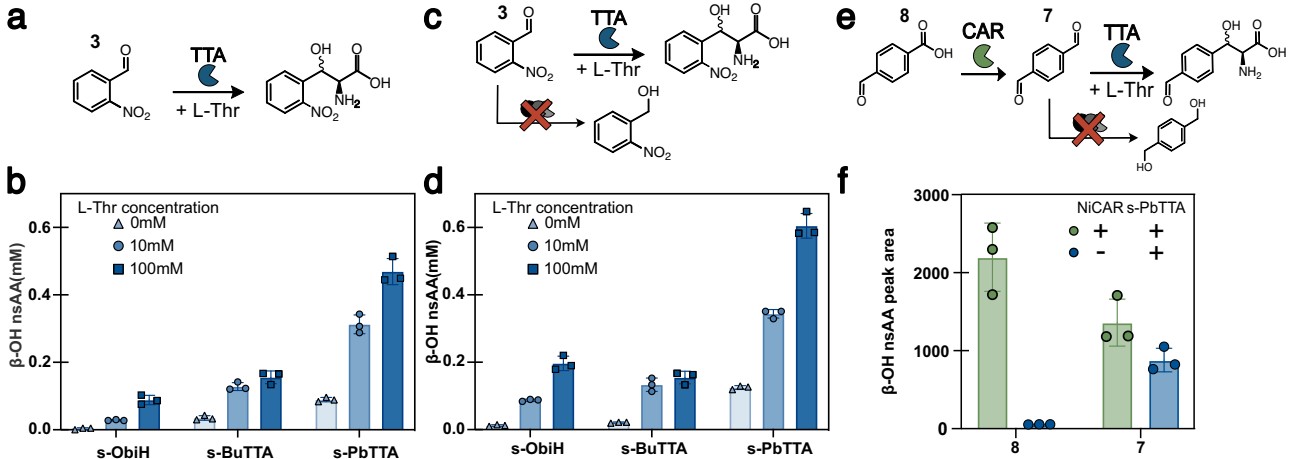

**Fig. 5 Biosynthesis of β-OH-nsAAs in metabolically active cells during aerobic fermentation. a** Schematic of β-OH-nsAA biosynthesis with supplemented aldehyde in a wild-type *E. coli* strain. **b** β-OH-nsAA titer measured after 20 h for s-ObiH, s-BuTTA, and s-PbTTA with 0, 10, and 100 mM of L-Thr supplemented and 1 mM **3** supplemented. **c** Schematic of β-OH-nsAA biosynthesis with genomic modifications to improve aldehyde stabilization. **d** β-OH-nsAA titer measured after 20 h for s-ObiH, s-BuTTA, and s-PbTTA with 0, 10, and 100 mM of L-Thr supplemented and 1 mM **3** supplemented. **e** Schematic of biosynthesis of β-OH-nsAA from an acid precursor when the TTA is coupled with a CAR in the RARE strain. **f** β-OH-nsAA peak area for 4-formyl-β-OH-phenylalanine from 4-formyl benzoic acid (**8**) and terephthalaldehyde (**7**) within the RARE strain with pACYC-NiCAR and pZE-s-PbTTA for the coupled production and RARE with pACYC-s-PbTTA, otherwise. Peak area is calculated as the area under the curve for the new peak corresponding to the product in the absorbance spectra for the appropriate wavelength from HPLC. All experiments were performed with technical triplicates. Each replicate is represented as its own data point with error bars representing standard deviations.

region for all the clusters containing TTAs. It appears that this region is a defining characteristic for many of these clusters (Fig. S30). Each cluster appears to have on average a different length which may contribute to distinct substrate specificities.

**In vivo production of β-OH-nsAAs.** Our last objective was to explore the biosynthesis of β-OH-nsAAs in metabolically active cells growing in aerobic conditions given our eventual desire to couple these products to ribosomal and non-ribosomal peptide formation. Production of the targeted β-OH-nsAA using cells that are growing during aerobic fermentation would need to meet three requirements: (1) Soluble expression of TTAs; (2) Affinity towards L-Thr at physiologically relevant concentration; (3) Import and stability of aryl aldehyde substrates in the presence of live cells. We hypothesized that the identified TTAs may perform better than ObiH in growing cells because the faster reaction rate of the enzyme could enable aldehyde utilization prior to aldehyde degradation by the cell. In addition, a higher affinity for L-Thr could improve titers achieved in the absence of supplemented L-Thr. Thus, we decided to test the top-performing TTAs in live cells and compare titers for different enzymes, specifically ObiH which has the highest expression, PbTTA which has the lowest L-Thr $K_M$ and highest $k_{cat}$ but low expression, and BuTTA which has the second highest catalytic rate with high expression. Using the SUMO-tagged constructs, each enzyme was screened in a 96-well plate, fermentative conditions in wild-type *E. coli* MG1655 with supplementation of either 0 mM, 10 mM, or 100 mM L-Thr and 1 mM **3**. We then analyzed titers after 20 h via HPLC analysis using the chemically synthesized β-OH-nsAA standard for the assumed product from **3**. PbTTA performed the best with the highest titer of $0.47 \pm 0.04$ mM β-OH-nsAA with 100 mM L-Thr supplemented as well as the highest titer with physiological levels of L-Thr at $0.09 \pm 0.01$ mM β-OH-nsAA in growing cells (Fig. 5a, b). Thus, we confirmed the production of the β-OH-nsAA in growing cell cultures; however, we wondered whether we could improve titer by implementing an aldehyde-stabilizing strain.

To investigate whether the knockout of genes that encode aldehyde reductases would result in improved yields of β-OH-

nsAA, we transformed the plasmid that harbors our TTA expression cassette into another *E. coli* strain that was engineered to stabilize aromatic aldehydes, the RARE strain[46]. The RARE strain has been shown to stabilize many aryl aldehydes, including **1**, **9**, and **12**, by eliminating potential reduction pathways[46,47]. We then repeated the experiment in the RARE strain and once again found that PbTTA produced the highest titer with $0.61 \pm 0.04$ mM produced with 100 mM L-Thr and $0.13 \pm 0.01$ mM produced with natural L-Thr levels (Fig. 5c, d). These 1.3x improvements in titer with the RARE strain suggest that stabilization of the aldehyde can improve β-OH-nsAA titers for certain chemistries, despite observing some reduction of the aldehyde to the corresponding 2-nitro-benzyl alcohol as well as reduction of the nitro-group (Fig. S31). While we did not see as large of an improvement for this chemistry as anticipated, our study suggests that the *E. coli* RARE strain transformed to express PbTTA is a promising chassis for β-OH-nsAA production during aerobic fermentation.

Finally, to partially address the toxicity of supplemented aldehydes in fermentative contexts, we investigated whether we could couple a TTA to a CAR to create a steady and low-level supply of aldehydes biosynthesized from carboxylic acid precursors. We coupled PbTTA to a well-studied CAR from *Nocardia iowensis* (NiCAR) to produce a β-OH-nsAA from the corresponding acid in aerobically growing RARE. We performed an initial screen with 2 mM 4-formyl benzoic acid (**8**), a proven substrate for NiCAR[48] but not for PbTTA, which would install a conjugatable aldehyde group onto a potential β-OH-nsAA product. We sampled cultures for HPLC analysis 20 h after the addition of the carboxylic acid precursor and observed a peak corresponding to the β-OH-nsAA (Fig. 5e, f). Additionally, there was greater production of the β-OH-nsAA when starting with the corresponding acid precursor compared to the aldehyde substrate, demonstrating that the addition of the CAR can improve final titers. To our knowledge, we are the first to demonstrate the production of this β-OH-nsAA from either the acid or the aldehyde and we were able to produce it in aerobically growing cells. Additionally, the RARE host maintains the aldehyde

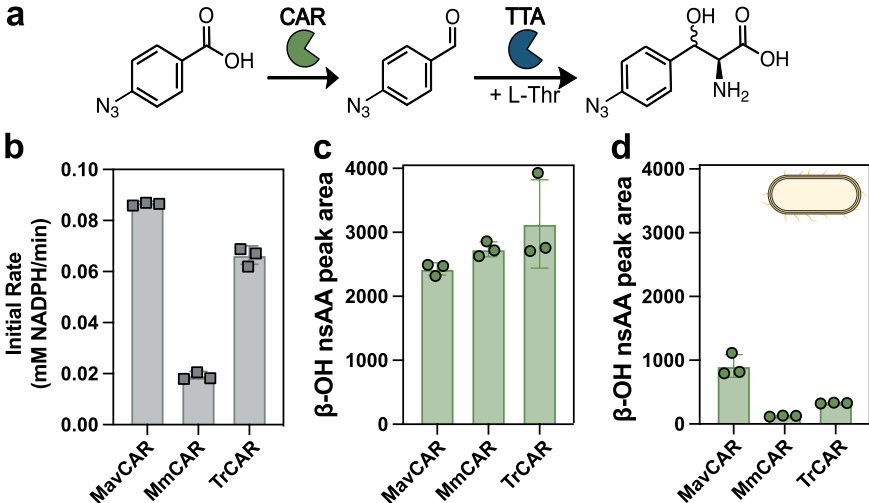

**Fig. 6 Activity of CARs and PbTTA to produce 4-azido-β-OH-phenylalanine. a** Reaction scheme for the conversion of 4-azido-benzoic acid to 4-azido-β-OH-phenylalanine. **b** Initial rate of NADPH depletion measured for three purified CARs when provided the previously unreported candidate substrate of 4-azido-benzoic acid. **c** β-OH-nsAA production measured by peak area for an in vitro coupled assay with the specified CAR and PbTTA. **d** β-OH-nsAA production measured by peak area in aerobically cultivated cells of the E. coli RARE strain transformed to express each CAR on a pZE vector and pACYC-s-PbTTA. Cultures were supplemented with 4-azido-benzoic acid during the mid-exponential phase and sampled after 20 h of growth. Peak area is calculated as the area under the curve for the new peak corresponding to the product in the absorbance spectra for the appropriate wavelength from HPLC. Experiments were performed in technical triplicate with each replicate represented. Error bars are standard deviations.

functional handle of the β-OH-nsAA. The addition of a CAR to this cascade limits the impact of aldehyde toxicity and instability on final product titers and provides the opportunity for future β-OH-nsAA production as a de novo pathway from glucose given the natural abundance of carboxylic acids.

**Pathway development for a bioconjugatable β-OH-nsAA**. With the promise of the CAR-TTA coupling, we wanted to investigate the generalizability of this pathway to produce a β-OH-nsAA that has a more common and versatile bio-orthogonal conjugation handle. We chose the 4-azido functionality as our target and explored whether it could be made from a 4-azido-benzoic acid precursor. To our knowledge, this precursor would be a substrate never previously tested with any CAR enzyme and its product would be a substrate never tested with any TTA enzyme. Given the prevalence of the azide group as a bio-orthogonal conjugation handle, we selected 4-azido-benzoic acid as the target substrate to produce the corresponding β-OH-nsAA product (Fig. 6a). We first studied a panel of three CARs with a diverse substrate scope and high soluble expression[48] (Fig. 6b). We observed activity of all the CARs on the acid substrate, so we then coupled the CAR directly to PbTTA in an in vitro assay to identify the β-OH-nsAA (Fig. 6c). The CAR-TTA coupling is valuable because the carboxylic acid precursor is 100-fold less costly to purchase than the aldehyde precursor and the aldehyde is likely to be toxic to cells if supplied at high concentrations. The in vitro coupling also successfully produced a β-OH-nsAA product verified as a new peak on the HPLC (Fig. S32). We did observe similar production across all CAR-TTA pairings despite the distinct activity of the CARs which suggests that PbTTA may be a limiting step in this cascade. Finally, given the potential to produce novel peptide or protein products in cells, we wanted to confirm the activity of this cascade in growing cells, which was successful for all CAR-TTA pairings with MavCAR producing the highest titer determined by product peak area after 20 h (Fig. 6d). To our knowledge, we are the first to produce a β-OH-nsAA that contains an azide functionality from either carboxylic acid or aldehyde precursors, which could be useful for chemical diversification of β-OH-

nsAAs, and associated products formed by fermentation using engineered bacteria.

## Discussion
We sought to expand the fundamental understanding of the TTA enzyme class to ultimately develop a platform E. coli strain for fermentative biosynthesis of diverse β-OH-nsAA from supplemented aromatic aldehydes or carboxylic acids. To achieve this, we had to overcome a series of challenges including low protein solubility, low activity on non-ideal substrates, and low L-Thr affinity. We successfully identified a solubility tag that improved the expression of 11 of the selected TTAs. We then expressed, purified, and tested nine previously uncharacterized enzymes at the study's outset. We successfully identified these TTAs through bioprospecting and rapid analysis of diverse enzymes via an in vitro TTA-ADH coupled assay. Of these characterized enzymes, we identified PbTTA, which expresses well in E. coli, can act on a diverse array of substrates, has a higher affinity towards L-Thr than ObiH, and has a higher catalytic rate when using 14 and L-Thr as substrates. We tested this enzyme in a series of fermentative contexts in an aldehyde-stabilizing strain and coupled it with a CAR to produce β-OH-nsAAs in aerobically grown cells.

Heterologous expression in model bacteria such as E. coli is a well-documented problem for many TTAs, including LipK, and FTase[19,31], where ObiH is the exception. The SUMO-tag appeared to improve the solubility of many enzymes that share sequence similarity to ObiH, LipK, and FTase, such that some enzymes that were unable to be expressed initially were expressed and purified. Fortunately, the SUMO-tag did not appear to impact enzyme activity for the enzymes screened, which agrees with predicted structures. Our findings and further computational predictions suggest that an N-terminal SUMO-tag may improve protein expression for similar sequences. Furthermore, our construct design facilitates the removal of the tag if needed without impacting enzyme structure.

As a target enzyme for broad amino acid biosynthesis, several studies of PsLTTA and ObiH suggest a trend of limited activity on aldehydes with electron-donating ring substituents and varying activity based on the position of the ring substitution[23–27]. We

observed similar trends with ObiH; however, we were able to expand the substrate scope to a variety of other substrates including those with some electron-donating properties like 4-methoxy-benzaldehyde, **9**. We identified substrates with amine chemistry that appeared to be substrates for ObiH, offering an opportunity for diversification of the potential β-OH-nsAA products. Other chemistries like 4-formyl-boronic acid, **13**, and terephthalaldehyde, **7**, can act as bioconjugatable and reactive handles for antibiotic and non-ribosomal peptide diversification, as well as for protein engineering applications. Additionally, we wanted to determine if these trends hold for the TTAs we identified. Using a selection of aldehydes with different electronic properties, we observed that the TTAs within the ObiH cluster (PiTTA, CsTTA, and BuTTA) maintain the trends observed with ObiH. Further, we observed that PbTTA has a broader substrate scope and maintains high activity on most substrates screened, including 4-azido-benzaldehyde produced from CAR coupling.

The combination of our SSN, our experiments, and our analysis using biosynthetic gene cluster (BGC) discovery tools[43] has revealed that TTAs may be much more versatile in the biosynthesis of natural or unnatural antibiotics than previously understood. The diversity of enzymes that we observed that had TTA activity suggests that there are likely many more natural enzymes capable of performing these aldol-like condensations. Additionally, the origin of ObiH, LipK, and FTase in natural product synthesis suggests that there may be other natural product syntheses that rely on this chemistry. For example, within the LipK-like enzyme cluster, there are eight published enzymes reported to be a part of several distinct nucleoside antibiotic biosynthetic gene clusters (Fig. S33). Of the enzymes we evaluated in our study, RaTTA and SNTTA are a part of predicted spicamycin and muraymycin BGCs, respectively (Table S2)[43,49]. Even with the addition of the SUMO-tag, we were only able to purify SNTTA and we observed no TTA activity on aromatic aldehydes. KaTTA, one of the active TTAs we identified, is a part of predicted valclavam BGC (Table S2). Upon further analysis, we identified OrfA and an OrfA-like protein described in the literature[50,51] that are in the same cluster as KaTTA. Interestingly, several enzymes tested and identified to have TTA activity are not a part of any known or characterized BGCs (BuTTA, PbTTA, StTTA-Δ36). This could provide an opportunity for further exploration of natural products based on the discovery of enzymes with this activity. BuTTA and PbTTA are two such enzymes that warrant further investigation into their genomic context for the elucidation of potential natural products.

Finally, we successfully developed an _E. coli_ strain for β-OH-nsAA production by using an aldehyde-stabilizing strain[46] and by coupling the TTA with a CAR for β-OH-nsAA production from an acid substrate. There are ample opportunities to explore additional aldehyde and acid substrates, develop new pathways from glucose, and improve accessible L-Thr concentrations with metabolic and genome engineering[52]. The production of diverse β-OH-nsAA in fermentative contexts should also enable the formation of complex ribosomally and non-ribosomally translated polypeptides for potential drug discovery. Ultimately, this study brings us a step closer to a platform _E. coli_ strain for the production of diverse β-OH-nsAAs in fermentative contexts.

## Methods

**Strains and plasmids**. _Escherichia coli_ strains and plasmids used are listed in Table S3. Molecular cloning and vector propagation were performed in DH5α. Polymerase chain reaction (PCR) based DNA replication was performed using KOD XTREME Hot Start Polymerase for plasmid backbones or using KOD Hot Start Polymerase otherwise. Cloning was performed using Gibson Assembly with constructs and oligos for PCR amplification shown in Table S4. Genes were purchased as G-Blocks or gene fragments from Integrated DNA Technologies (IDT) or Twist Bioscience and were optimized for _E. coli_ K12 using the IDT Codon Optimization Tool with sequences shown in Table S5. The following plasmids are available on Addgene with the Addgene ID listed in parentheses: P14 (204629), P15 (204630), P17 (204631), P18 (204632), P24 (204633), and P25 (204634).

**Materials and chemicals**. The following compounds were purchased from MilliporeSigma (Burlington, MA, USA): kanamycin sulfate, dimethyl sulfoxide (DMSO), potassium phosphate dibasic, potassium phosphate monobasic, magnesium chloride, calcium chloride dihydrate, imidazole, glycerol, beta-mercaptoethanol, sodium dodecyl-sulfate, lithium hydroxide, boric acid, Tris base, glycine, HEPES, L-threonine, L-serine, adenosine 5′-triphosphate disodium salt hydrate, pyridoxal 5'-phosphate hydrate, benzaldehyde, 4-nitro-benzaldehyde, 4-amine-methyl-benzaldehyde, 4-formyl benzoic acid, 4-methoxybenzaldehyde, 2-naphthaldehyde, 4-formyl boronic acid, NADH, phosphite, Boc-glycine-OH, trimethylacetyl chloride, (1R,2R)-2-(Methylamino)-1,2-diphenylethanol, trifluoroacetic acid, alcohol dehydrogenase from _S. cerevisiae_, and KOD XTREME Hot Start and KOD Hot Start polymerases. Lithium bis(trimethylsilyl)amide, 4-dimethyl-amino-benzaldehyde, and 2-amino-benzaldehyde were purchased from Acros (Geel, Belgium). D-glucose, 2-nitro-benzaldehyde, 4-biphenyl-carboxaldehyde, terephthalaldehyde, and 4-azido-benzoic acid were purchased from TCI America (Portland, OR, USA). Agarose, Laemmli SDS sample reducing buffer, 4-tert-butyl-benzaldehyde, phenylacetaldehyde, and ethanol were purchased from Alfa Aesar (Ward Hill, MA, USA). 2-nitro-phenylacetaldehyde and 4-nitro-phenylacetaldehyde were purchased from Advanced Chem Block (Burlingame, CA, USA). Anhydrotetracycline (aTc) was purchased from Cayman Chemical (Ann Arbor, MI, USA). Hydrochloric acid was purchased from RICCA (Arlington, TX, USA). Acetonitrile, methanol, sodium chloride, LB Broth powder (Lennox), LB Agar powder (Lennox), Amersham ECL Prime chemiluminescent detection reagent, bromophenol blue, and Thermo Scientific™ Spectra™ Multicolor Broad Range Protein Ladder were purchased from Fisher Chemical (Hampton, NH, USA). NADPH was purchased through ChemCruz (Dallas, TX, USA). A MOPS EZ-rich defined medium kit and components were purchased from Teknova (Hollister, CA, USA). Trace Elements A was purchased from Corning (Corning, NY, USA). Taq DNA ligase was purchased from GoldBio (St. Louis, MO, USA). Phusion DNA polymerase and T5 exonuclease were purchased from New England BioLabs (NEB) (Ipswich, MA, USA). Sybr Safe DNA gel stain was purchased from Invitrogen (Waltham, MA, USA). HRP-conjugated 6*His His-Tag Mouse McAB was obtained from Proteintech (Rosemont, IL, USA).

**Overexpression and purification of threonine transaldolases**. A strain of _E. coli_ BL21 transformed with a pZE plasmid encoding expression of a TTA with a hexahistidine tag or a hexahistidine-SUMO tag at the N-terminus (P1-P26) was inoculated from frozen stocks and grown overnight in 5 mL LBL containing kanamycin (50 μg/mL). Overnight cultures were used to inoculate 250–400 mL of experimental culture of LBL supplemented with kanamycin (50 μg/mL). The culture was incubated at 37 °C until an $OD_{600}$ of 0.5–0.8 was reached while in a shaking incubator at 250 RPM. TTA expression was induced by the addition of anhydrotetracycline (aTc) (0.2 μM) and cultures were incubated shaking at 250 RPM at either 18 °C for 24 h, 30 °C for 5 h then 18 °C for 20 h or 30 °C for 24 h. Cells were centrifuged using an

Avanti J-15R refrigerated Beckman Coulter centrifuge at 4 °C at $4000 \times g$ for 15 min. The supernatant was then aspirated and pellets were resuspended in 8 mL of lysis buffer (25 mM HEPES, 10 mM imidazole, 300 mM NaCl, 400 μM PLP, 10% glycerol, pH 7.4) and disrupted via sonication using a QSonica Q125 sonicator with cycles of 5 s at 75% amplitude and 10 s off for 5 min. The lysate was distributed into microcentrifuge tubes and centrifuged for 1 h at 18,213 x g at 4 °C. The protein-containing supernatant was then removed and loaded into a HisTrap Ni-NTA column using an ÄKTA Pure GE FPLC system. Protein was washed with 3 column volumes (CV) at 60 mM imidazole and 4 CV at 90 mM imidazole. TTA was eluted in 250 mM imidazole in 1.5 mL fractions over 6 CV. Samples from selected fractions were denatured in Lamelli SDS reducing sample buffer (62.5 mM Tris-HCl, 1.5% SDS, 8.3% glycerol, 1.5% beta-mercaptoethanol, 0.005% bromophenol blue) for 10 min at 95 °C and subsequently run on a sodium dodecyl-sulfate polyacrylamide gel electrophoresis (SDS-PAGE) gel with a Thermo Scientific PageRuler™ Prestained Plus ladder to identify protein-containing fractions and confirm their size. The TTA-containing fractions were combined and applied to an Amicon column (10 kDa MWCO) and the buffer was diluted 1000x into a 25 mM HEPES, 400 μM PLP, 10% glycerol buffer. This same method was used for purification of the CAR enzymes, *E. coli* pyrophosphatase, *E. coli* ADHs, and phosphite dehydrogenase (PTDH).

**Threonine transaldolase expression testing**. To test expression of the threonine transaldolase library, 5 mL cultures of MAJ14-26 and MAJ53-65 were inoculated in 5 mL cultures of LBL containing 50 μg/mL kanamycin and then grown shaking at 250 RPM at 37 °C until mid-exponential phase (OD = 0.5–0.8). At this time, cultures were induced via the addition of 0.2 μM aTc and then grown shaking at 250 RPM at 30 °C for 24 h. After this time, 1 mL of cells was mixed with 0.05 mL of glass beads and then vortexed using a Vortex Genie 2 for 15 min. After this time, the lysate was centrifuged at $18,213 \times g$ at 4 °C for 30 min. Lysate was denatured as described for the overexpression and then subsequently run on an SDS-PAGE gel with Thermo Scientific™ Spectra™ Multicolor Broad Range Protein Ladder and then analyzed via western blot with an HRP-conjugated 6*His, His-Tag Mouse McAB primary antibody. The blot was visualized using an Amersham ECL Prime chemiluminescent detection reagent.

**In vitro enzyme activity assay—TTA-ADH coupled enzyme assay**. High-throughput screening of purified TTAs was performed with a TTA-ADH coupled assay using purified TTA and commercially available alcohol dehydrogenase from *S. cerevisiae* purchased from Millipore Sigma. Aldehyde stocks were prepared in 50–100 mM solutions in DMSO or acetonitrile. Reaction mixtures were prepared in a 96-well plate with 100 μL of 100 mM phosphate buffer pH 7.5, 0.5 mM NADH, 0.4 mM PLP, 15 mM MgCl₂, and 100 mM L-Thr with the addition of 0.25 mM to 1 mM aldehyde depending on the background absorbance at 340 nm (Table S1), 10 U ScADH, and 0.25 μM purified TTA unless otherwise specified. Reactions were initiated with the addition of an enzyme. Reaction kinetics were observed for 20–60 min in a SpectraMax i3x microplate reader at 30 °C with 5 s of shaking between reads with the high orbital shake setting. The following controls were included for every assay: reaction mixture without aldehyde, without TTA, and without enzyme (TTA or ADH). Rates were calculated by identifying the linear region at the beginning of the kinetic run and converting the depletion in absorbance to the depletion of mM NADH using an NADH standard curve.

**In vitro enzyme activity assay—CAR-TTA coupled enzyme assay**. In vitro CAR activity assays were performed as previously reported[48] using 2 mM NADPH and 2 mM ATP, 20 mM MgCl₂, and 0.75 μM CAR and *E. coli* pyrophosphatase. For in vitro coupling with the CAR and TTA, the same in vitro CAR assay was performed with the addition of 2 μM TTA, 0.4 mM PLP, and 100 mM L-Thr; however, rather than monitoring the reaction with the plate reader, the plate was left shaking at 1000 RPM with an orbital radius of 1.25 mm at 30 °C overnight. The reaction was then quenched after 20 h with 100 μL of 3:1 methanol:2 M HCl. The supernatant was then separated from the protein precipitate using centrifugation and analyzed via HPLC.

**HPLC analysis**. Metabolites of interest were quantified via HPLC using an Agilent 1260 Infinity model equipped with a Zorbax Eclipse Plus-C18 column. To quantify aldehyde and β-OH-nsAAs, an initial mobile phase of solvent $A/B = 95/5$ was used (solvent A, water + 0.1% TFA; solvent B, acetonitrile + 0.1% TFA) and maintained for 5 min. A gradient elution was performed (A/B) as follows: gradient from 95/5 to 50/50 for 5–12 min, gradient from 50/50 to 0/100 for 12–13 min, and gradient from 0/100 to 95/5 for 13–14 min. A flow rate of $1 \, mL \, min^{-1}$ was maintained, and absorbance was monitored at 210, 250, and 280 nm.

**Culture conditions**. For screening TTA activity in aerobically growing cells, we inoculated strains transformed with plasmids expressing TTAs into 300 μL volumes of MOPS EZ Rich media in a 96-deep-well plate with appropriate antibiotic added to maintain plasmids (50 μg/mL kanamycin (Kan)). Cultures were incubated at 37 °C with shaking at 1000 RPM and an orbital radius of 1.25 mm until an $OD_{600}$ of 0.5–0.8 was reached. $OD_{600}$ was measured using a SpectraMax i3x plate reader. At this point, the TTAs were induced with the addition of 0.2 μM aTc for TTA expression. Then, 2 h following induction of the TTAs, 1 mM aldehyde was added to the culture. Cultures were then incubated for 20 h at 30 °C with metabolite concentration measured via supernatant sampling and submission to HPLC.

For the CAR-TTA coupled assay, the strains transformed with a plasmid expressing a TTA and a second plasmid expressing a CAR were grown under identical conditions with the addition of 34 μg/mL chloramphenicol (Cm) to maintain the additional plasmid. Further, 0.2 μM aTc and 1 mM IPTG were added to induce protein expression, and 2 mM aldehyde, or acid was added at the time of induction. Following induction, the cultures were grown for 20 h at 30 °C while shaking at 1000 RPM with product concentrations measured via supernatant sampling and submission to HPLC.

**Creation of Protein Sequence Similarity Network (SSN)**. Using NCBI BLAST, the 500 most closely related sequences as measured by BLASTp alignment score were obtained from three characterized threonine transaldolases, FTase, LipK, and ObiH. After deleting duplicate sequences[53], 1195 unique sequences were obtained, which were then submitted to the Enzyme Function Initiative-Enzyme Similarity Tool (EFI-EST)[50] to generate a sequence similarity network (SSN). Sequences exhibiting greater than 95% similarity were grouped into single nodes, resulting in 859 unique nodes, and a minimum alignment score of 85 was selected for node edges. The SSN was visualized and labeled in Cytoscape[54] using the yFiles Organic Layout.

**Sequence alignment**. Multiple sequence alignments were performed using ClustalOmega alignment within JalView[55] using the "dealign" setting and otherwise default settings of one for max

guide tree iterations, and one for the number of iterations (combined). The sequence identity matrix was generated using the online interface for the Multiple Sequence Alignment tool from ClustalOmega[38].

**Structure prediction.** Structures of the putative TTAs were produced using AlphaFold2 CoLab notebook[56] using the provided default settings with no template, the MMseqs2 (UniRef +Environmental) for multi-sequence alignment, unpaired +paired mode, auto for model_type and 3 for num_recycles. We then moved forward with the model ranked the highest. We performed the alignment of chains A and B from the crystal structure of ObiH (PDB ID: 7K34) and the AlphaFold model for PbTTA using the align command in PyMOL with all default settings. The same alignment protocol was implemented for aligning the AlphaFold2 models of putative TTAs with and without the SUMO tag.

**Statistics and reproducibility.** All experiments were performed in either biological or technical triplicate as reported in the figure legends. Triplicate data is represented in each figure as separate data points and error bars represent standard deviations across triplicate values. Some full experiments were performed in replicates on separate days to confirm reproducibility. The data represented within the manuscript is that of a single day's experiments that is representative of all replicates.

**Reporting summary.** Further information on research design is available in the Nature Portfolio Reporting Summary linked to this article.

## Data availability

The datasets generated during and/or analyzed during the current study are contained in the published article (and its Supplementary Information) and are available from the corresponding author upon reasonable request. Source data for figures can be found in Supplementary Data 1.

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

## Acknowledgements

We are grateful to Jessica Sampson and Cameron Twitty of the University of Delaware High-Throughput Experimentation Facility for assistance with metabolite LC-MS. We are also grateful to Sabyasachi Sen, Madan Gopal, Roman Dickey, and Priyanka Nain for providing genetic constructs and purified proteins. We acknowledge support from the following funding sources: The Office of Naval Research Award No. N000142212536 (to A.M.K.); the Department of Education—Graduate Assistance in Areas of National Need P200A210065 (to M.A.J.); the National Institute of General Medical Sciences of the National Institutes of Health under a Chemistry-Biology Interface Training Grant T32GM133395 (to M.A.J. and S.A.); the National Science Foundation Collaborative Research Grant MCB-2027074 (to A.M.K.); and the National Institute of General Medical Sciences of the National Institutes of Health under Award Number P20GM104316.

## Author contributions

A.M.K. conceived and supervised the study; M.A.J. designed and performed most experiments analyzed data, prepared figures, and wrote the manuscript; N.D.B. and S.A.W. generated the SSN, selected the TTAs, cloned the TTAs, tested expression, and purified TTAs; S.R.A. performed the in vivo CAR-TTA coupling experiments; I.G. aided with optimizing the TTA-ADH coupled assay; X.L. and Y.F. performed organic synthesis of the chemical standard.

## Competing interests

The authors declare the following competing interests: M.A.J., S.A.W., N.D.B., and A.M.K. are co-inventors on a filed patent application related to this work. N.D.B. and A.M.K. have a financial interest in a commercial entity, Nitro Biosciences Inc. A.M.K. is a member of the scientific advisory board of Wild Microbes. The remaining authors declare no competing interests.
