## [Peer Review File · Communications Biology]

Reviewers' comments:

Reviewer #1 (Remarks to the Author):

The application of an enzyme-catalyzed aldol reaction provides a stereoselective pathway to β -hydroxy- α -amino acids using aldehyde and amino acid. In especial, L-Threonine aldolases (TTAs) constitute a powerful tool for catalyzing C-C bond formations in synthetic organic chemistry. Here, the authors ascertained the specificity of the TTAs enzyme class more comprehensively by characterizing 12 candidate TTAs gene products across a wide range of sequence identities. They found that the addition of a solubility tag substantially enhanced the soluble protein expression level within this difficult-to-express enzyme family. This work reveals an unexpectedly high level of sequence diversity and broad substrate specificity in an enzyme family whose members play key roles in the biosynthesis of therapeutic natural products. I think this work is a pretty important complement to the research in the field of TTAs. It has the high level of novelty and synthetic significance required for Communication Biology. Therefore, it would be suitable for publication in this journal, if the following issues could be addressed.

1: The author shows a broad substrate. But I am still curious about some other substrates which also been reported in other papers. I don't know if the author can do it?

2. I'm very curious about the d.e. (syn: anti) of the threonine analogs. It's very important in organic chemistry. As far as I know, most of the selectivity in previous studies of TTAs enzymes was not very good. I know this may be beyond the scope of this work. But I still think it's necessary to show some representative result.

3. I notice the author did not show the yield or TTN. I'm not very sure if it's acceptable for Communication Biology.

4. Generally, I find the citations given to previous work sufficient. Including the work by Wencewicz (2017) and Wilkinson (2017). I think the two citations (and maybe more) of using the ObiH for biocatalytic cascades (e.g. Andrew, *Angew. Chem. Int. Ed.* 2022, 61, e202212637) should also be included in the introduction.

Except for the few points raised above, I find the manuscript well-written and clear. The logic is very clear and precise. In my view, a revised version of the manuscript might very well be publishable in Communication Biology.

Reviewer #2 (Remarks to the Author):

Nonstandard amino acids are important building blocks for the production of small molecule antibiotics and polypeptides. Therefore, harnessing the enzymatic machinery responsible for their biosynthesis is of great interest for applications in biocatalysis and, more in general, biotechnology.

In this work, Jones and coworkers set out to discover and characterize new L-Threonine Transaldolases (TTAs), which are able to perform aldol condensation reactions between L-Thr and different types of aldehydes to yield nonstandard β -OH amino acids (β -OH-nsAA). By using a multidisciplinary approach—involving computational methods such as network analysis and protein sequence/structure alignments, as well as synthetic biology tools, enzymatic assays, and analytical chemistry techniques—the authors successfully identify, express, and characterize several catalytically active TTAs distantly related to each other. Furthermore, they establish an elegant in vivo system that can be used for the production of β -OH-nsAA (with functional handles) starting from accessible substrates such as L-Thr and carboxylic acids.

In my opinion, the research described here is of great quality and the rationale behind the experiments is solid. All the proper controls have been carried out and all experiments are presented

in triplicates in a visually compelling manner. The extensive supplementary material file demonstrates that the authors approached this work thoroughly and it provides a lot of useful information to go along with the main manuscript, especially should other researchers wish to replicate some of the work. In general, I believe that the results presented in this work can lead to the development of efficient biocatalytic processes for important building blocks. Lastly, the approach used in this work can benefit other researchers who wish to discover and study other types of biocatalysts.

I only have a few points that I would like to address:

1. Lines 85-86 & Fig. 1: the authors state that they tackle specific challenges related to in vivo biosynthesis of β -OH-nsAA, including aldehyde substrate stability, and they then refer to Fig. 1c. From this figure it is not really clear to me that this particular problem is addressed (as it is instead in Fig. 5c). Maybe the figure can be adjusted accordingly.

2. Lines 186-187: why is phenylacetaldehyde being used for kinetics, when one of the best performing substrates for these enzymes and assay is 2-nitrobenzaldehyde (lines 174-175, Fig. 2 and 5)? This should be explained further.

3. Lines 236-237: the authors state that KaTTA has a different residue in place of Y55 (ObiH numbering), but this is not what is shown in the multiple sequence alignment in Fig. S27. There it is clear that KaTTA also has a Y residue (position 96 relative to the MSA), so is this a mistake or did the authors come to this conclusion from a different analysis? This should be corrected or clarified.

4. Fig. S27: Something appears to have gone wrong with the processing of this figure: in the second panel (positions 60-119), the residues in the very last column are not aligned and seem to overlap in some positions.

5. Line 227: Why was NoTTA not tested with other aldehyde substrates? Is there a specific reason? Clarify if possible.

6. Lines 245-246: I wonder what the rationale behind this statement is. Admittedly, I am not an expert in structural biology, but I assume that the rigid structure in PbTTA would make for a stricter substrate specificity. Which is the opposite of what the authors speculate. Maybe they can expand their thoughts further.

7. Line 252: This is connected with my 3rd comment. Why do the authors say that KaTTA does not have a residue aligned to Y55? The MSA shows it quite clearly, and if you have reached this conclusions through other analyses (e.g. structural alignment) you should clearly state and show that.

8. Fig. S28: Along the same lines as the comment above, here the authors report "no alignment" for the KaTTA cluster with respect to the residue Y55. In accordance with comments 3 and 7, this should be clarified.

9. Lines 419-420: the use of the terms "confluence" and "confluent" is technically not correct, as it only applies to cultures that grow as a monolayer when they cover the entire adhesive surface of their culturing vessel (e.g. most types of mammalian cells). For freshly saturated bacterial cultures grown in suspension, "overnight" alone is sufficient.

All in all, I would like to thank the authors for the very enjoyable read, and I am very much looking forward to seeing the final version of this manuscript.

Dr. Riccardo Iacovelli

Reviewer #3 (Remarks to the Author):

A. Summary of key results: In this work the authors discover, characterize, and develop fermentation strategies for making various non-standard beta-hydroxylated amino acids. Authors first characterized promiscuous activity of threonine transaldolase enzyme, ObiH, and a variety of new aldehyde acceptors. Using ObiH as a starting point, they performed bioprospecting through SSNs and experimentally characterized a variety of novel TTA enzymes. Extending their results beyond biocatalysts, they showed that these enzymes can be made compatible with fermentative conditions for making beta-hydroxylated amino acids biosynthetically. Finally, they showed how all the developments from above can be used for in vivo biosynthesis of an azide-containing beta-hydroxy amino acid that has applications in genetic code expansion.

B. Originality and significance: Threonine transaldolases have recently been garnering high interest for many of the reasons highlighted in this work. In terms of significance and originality, the work here presented by the authors is the most extensive in both scope and breadth. It characterizes different 16 aldehyde acceptors across 13 different TTA enzymes (the majority of which have never been characterized before). Authors developed and validated a new coupled assay that made screening these 208 reaction conditions scalable. Importantly, quantitative kinetic data was collected for all these conditions that establish a rich dataset for future research into this class of enzymes. Where I believe this work goes beyond what others attempt is in extending results beyond biocatalysts and moving to fermentation. Authors show how fermentation using aldehyde building blocks can be made possible by using a specially engineered ADH-knockout strain. Realizing some of the shortcomings of feeding aldehydes, they cleverly show that carboxylic acids can serve as donors if coupled to overexpression of promiscuous CAR enzymes. The final result presented in this work, azido-amino acid biosynthesis, is original and interesting in its own respect for applications in genetic code expansion.

C. Data and methodology: Methods used throughout the paper were appropriate and presented in good quality. Data generated was clearly reproducible and experimental conditions used were well documented. High degree of rigor was present as all data was collected with replicates, was quantitative (if standards were present), and used orthogonal analytical tools for validation (HPLC, LC/MS).

D. Appropriate use of statistics and treatment of data. Statistics were correctly presented but there are minor suggestions regarding what is presented and some missing detail.

- Kinetics: The authors characterize kinetics of TTA enzymes but they should provide a point of clarity in methods or text regarding aldehyde component: (1) Specify if this was done at saturation for aldehyde, (2) explicitly specify it was done at a fixed aldehyde concentration and justify, or (3) provide literature precedence for aldol condensation of TTA enzyme reaction not being rate limiting.
- Kinetics: Authors need to specify what strategy was used for calculating MM parameters in the methods section.
- Kinetics: SI MM plots (S36) have incorrect units for initial rates. Should be $\mu\text{M}/\text{min}$ not mM/min .
- Kinetics: SI MM plots (S36) authors should specify concentration of enzyme used here OR plot rate/ E_0 instead of rate.
- Kinetics: 95% CI intervals are presented for MM parameters. I would suggest authors report standard deviation instead.

E. Conclusion: Conclusion presented is supported by evidence provided in the paper.

F. Suggested improvements:

- Page 10 Line 172-179: Consider rewriting this paragraph for better narrative.
- Figure S27 – Clipping on the right side, please fix.
- Provide residue number of homologues when talking about alignments. Can be tabulated in the SI. I had trouble finding which residues were being referred to in alignment since the alignments shown in SI for ObiH are numbered differently.

G. References: Appropriate credit to previous work was used.

H. Clarity and context: Overall clarity and context provided in manuscript is high. Narrative was easy to follow and results were properly contextualized with literature. Some comments that will need to be addressed are:

- Some paragraphs, noted in minor comment, could use improvements in clarity.
- In SI authors state: "For StTTA, the first 36 amino acids at the N-terminus were removed to improve the similarity between StTTA and ObiH." This statement needs to be moved into the main text and justified. It is not entirely obvious why and could help explain observations. StTTA should be shown in the main text as an Nt truncation (e.g., StTTA Δ 1-36 or something similar).
- Figure S5 in SI: LC/MS should be better annotated. Showing raw chromatogram is not super helpful. Other chromatograms in the SI do this properly and label known peaks clearly.

General comment:

Main text:

- Can authors comment (in a response) on these aspects of their assays and coupled assay:
 - o Absorbance of pyrodoxal 5 phosphate and intermediates
 - o Reduction of pyrodoxal 5 phosphate by aldehyde dehydrogenase
 - o Pyrodoxal 5 phosphate serving as an acceptor for aldol condensation

Figures:

- Figure 4: Suggest changing "initial rate" axis title name to Initial rate/E₀ (or similar) to distinguish from initial rate calculations that are not enzyme normalized.
- Figure 5, 6: Suggest changing "peak area" to whatever the measurement was instead OR specify in the figure legend that it is peak area as measured by X. Peak area is vague so it is hard to know what I am looking at.

Supplementary information:

- Table S1: What is the meaning of the E. coli strain number? If these numbers are never referenced in a meaningful way, I would suggest removing this column. Same with plasmid numbers.
- Table S2: Since most of the primers in this table are never explicitly referenced, can the authors more clearly describe what the primers are for? Cloning? Sequencing?
- Table S3: Protein accession number (column header) and accession numbers are wrapped. Since you are wrapping sequence, you can just make accession number column wider to avoid this.
- S11: "For StTTA, the first 36 amino acids at the N-terminus were removed to improve the similarity between StTTA and ObiH." Authors need to justify this more as it does not make sense to want to screen natural homologues but making large arbitrary deletions.
- S25: Caption title is missing a word "L-thr is required for ..."
- S28: In the main text you said that Y55 was aligned for KaTTA, but this disagrees with what is in S27 and in this figure (S28). I think there just needs to be a bit more clarity on the alignment and its meaning.

Minor comments:

Page 2 Line 21: Should "candidate TTA gene products" be "candidate tta gene products" since it refers to the TTA gene rather than protein?

Page 3 Line 34: Side note: Also found in ribosomal peptide natural products like ustiloxin B in case authors wanted an example.

Page 3 Line 35: Droxidopa should not be capitalized since it is not a brand name, rather a chemical name.

Page 3 Line 36: "Some of these molecules" ambiguous – referring to either the 3 examples stated previously or aromatic nsAAs in general. Use more precise language.

Page 3 Line 45: I think it would be wise to be mentioning both NRP and RiPPs are potential applications, otherwise it will appear like you miss a large natural product class. Though antimicrobial peptides are mentioned, RiPPs should be explicitly stated as these are a specific subclass of natural product.

Page 3 Line 46: Mentioning how beta hydroxy groups could be installed post translationally in a RiPP would be helpful for authors here as well (since this is another strategy they can be made that is not

mentioned).

Page 4 Line 52: Suggest remove "Fortunately"

Page 4 Line 54: Suggest revising this statement since it is technically incorrect "catalyze the aldol condensation of L-threonine (L-Thr) with an aldehyde". The aldol condensation occurs through a quinonoid intermediate after a retroaldol cleavage a threonyl-PLP intermediate.

Page 4 Line 42-63: The phrasing "act on" when describing transaldol reactions could be made more precise by specifying saying "aldehyde acceptors" instead.

Page 4 Line 66: Suggest adding commas here maybe "ObiH, and TTAs more broadly, "

Page 5 Line 77: The reported KM of ObiH for L-Thr

Page 5 Line 77: Consider revising sentence for clarity to more clearly state that it would not be compatible with fermentative conditions in E. coli where thr concentrations are low.

Page 5 Line 84: Please do not tackle, just address.

Page 6 Line 92: Suggest rewriting sentence to avoid possessive "Our best"

Page 9 Line 142: "We were excited by" suggest revising since it you could sound more objective in the value of your finding "These results have important implications for"

Page 9 Line 148: Can the authors specify here what they think the limitations of ObiH here are? As written, it is left to the reader to interpret what is missing.

Page 10 Line 167: "We were excited to observe" suggest revising

Page 10 Line 172: "we identified the putative TTAs with high activity" – Though mentioned later in the paragraph, authors need to specify earlier what their objective is since activity is going to be subjective towards the substrate they chose.

Page 10 Line 177: KaTTA with and without SUMO tag statement seems out of place (as written).

Leaves open the question of why you did not do all the enzymes?

Page 10 Line 172-179: I think the experiments performed in this paragraph are all fine, I think the narrative of the paragraph could use a bit of work.

Page 12 Line 212-227: This paragraph reads as a medly of additional experiments. I appreciate the reporting of the negative results, and want to suggest that authors could try to (1) tie the narrative of this paragraph better together or (2) highlight why the negative results are important.

Page 13 Line 236: Semicolon unnecessary, start new sentence.

Page 13 Line 236: "Many of the active TTAs within the ObiH cluster had the same residues at these sites; however, PbTTA and KaTTA appeared to have modified residues at Y55 and E107 which are reported to perform hydrogen bonding for PLP stabilization (Fig. 4d)". Your alignment (S27) shows KaTTA as having Y55.

Page 14 Line 244: If you are speculating here, please provide clarity as to what the structure-to-function link is.

Page 14 Line 247-257: I am not sure phenomenological discussion of conserved residues is a useful topic of discussion in the absence of functional characterization.

Page 15 Line 265: "Productivity" is not well defined here

Page 17 Line 315: "Expensive" is relative. I would refrain from attaching value to what a chemical supplier might charge. I would suggest

Page 22 Line 393: Consider changing to "Materials and chemicals" since not everything listed is a chemical.

Page 22 Line 393: It might be good practice to include supplier headquarters location to remove ambiguity.

Page 24 Line 443: I am fairly sure this is a typo: 0.2 nM aTc should be 0.2 μ M aTc.

Page 27 Line 486: I am fairly sure this is a typo: 0.2 nM aTc should be 0.2 μ M aTc.

Page 27 Line 491: I am fairly sure this is a typo: 0.2 nM aTc should be 0.2 μ M aTc.

Page 29 Line 519: I would just remove the word cartoon here, say "Depiction of..."

We appreciate all the reviewer feedback, which we considered thoughtful and constructive. In the point-by-point response below, we have included the original comments colored in black, our responses colored in purple, and modifications to the text colored in red.

Table of Contents

Reviewer #1:	1
Reviewer #2:	3
Reviewer #3:	6

Reviewers' comments:

Reviewer #1:

The application of an enzyme-catalyzed aldol reaction provides a stereoselective pathway to β -hydroxy- α -amino acids using aldehyde and amino acid. In especial, L-Threonine aldolases (TTAs) constitute a powerful tool for catalyzing C-C bond formations in synthetic organic chemistry. Here, the authors ascertained the specificity of the TTAs enzyme class more comprehensively by characterizing 12 candidate TTAs gene products across a wide range of sequence identities.

They found that the addition of a solubility tag substantially enhanced the soluble protein expression level within this difficult-to-express enzyme family. This work reveals an unexpectedly high level of sequence diversity and broad substrate specificity in an enzyme family whose members play key roles in the biosynthesis of therapeutic natural products. I think this work is a pretty important complement to the research in the field of TTAs. It has the high level of novelty and synthetic significance required for Communication Biology. Therefore, it would be suitable for publication in this journal if the following issues could be addressed.

1: The author shows a broad substrate. But I am still curious about some other substrates which also been reported in other papers. I don't know if the author can do it?

We thank the reviewer for their comments and curiosity about the substrates that they provided the chemical structures of during review. While we are interested in investigating TTA activity on additional candidate substrates in ongoing and future work, we have had to make a decision about what number and type would be suitable for this initial manuscript. We have already chosen to test 16 substrates against ObiH and another 7 substrates across a wide panel of enzymes, and this led to over 60 reactions. Ongoing work in our lab is examining one or two of the TTAs that exhibited the broadest substrate specificity in more detail and optimizing conditions for specific substrates, one of which the reviewer expressed interest in. As the ongoing work is more narrowly focused on certain TTAs and chemistries that have distinct applications, we consider it out of scope for this manuscript and intended for a subsequent publication. Given the reviewer's comment, we will happily include the additional suggested substrates in our follow-up manuscript. Additionally, we

will deposit strains that contain these plasmids on Addgene so that others can test additional substrates in parallel.

Line 411: “The following plasmids are available on Addgene with the Addgene ID listed in parentheses: P14 (204629), P15 (204630), P17 (204631), P18 (204632), P24 (204633), and P25 (204634).”

2. I’m very curious about the d.e. (syn: anti) of the threonine analogs. It’s very important in organic chemistry. As far as I know, most of the selectivity in previous studies of TTAs enzymes was not very good. I know this may be beyond the scope of this work. But I still think it’s necessary to show some representative result.

We agree with the reviewer that the de% for the final products is of high interest, and we have included this result as SI Figure 24. We calculated the de% for the 2-nitrobenzaldehyde product since that is the only substrate for which we were able to synthesize a pure chemical standard. We calculated and reported the de% in the SI for all of the active TTAs in the study towards this substrate. We were excited to find that two of the novel TTAs we identified have improved stereoselectivity for this substrate and hope that they can be very useful to the community. We have highlighted this result more clearly in the main text with the following adjustments:

Line 201: “Further, this result confirmed that after 20 h, ObiH, PiTTA, CsTTA, BuTTA, KaTTA, and PbTTA all approached 100% conversion of the aldehyde to the final β -OH-nsAA product. Also, KaTTA and PbTTA produce almost stereochemically pure isomers of the *threo* β -OH-nsAA with de% of 97% and 98%, respectively, which is better than the de% of 80% for products from ObiH (Fig. S24).”

3. I notice the author did not show the yield or TTN. I’m not very sure if it’s acceptable for Communication Biology.

We would like to thank the reviewer for this observation. Unfortunately, we are unable to quantify the yield or TTN for most of the substrates we produced because we lack a standard for quantification via HPLC or LC-MS. We also are unable to purify the compound using the systems available to us. We were able to calculate the percent of the aldehyde converted for an experiment using the β -OH nsAA standard produced from 2-nitro-benzaldehyde and have added this information to the SI Fig. 25 and highlighted the result in the text below. Rather than reporting yields or TTN, we were interested in determining initial enzyme activity and thus relied on the TTA-ADH coupled enzyme assay for a higher throughput screen of enzyme activity. We are confident in the results observed from this assay due to rigorous experimentation and validation of the assay itself. Further, we believe publications such as the discovery of the obafluorin pathway from Scott et al. in *Nature Communications*, the bioprospecting for hydroxyproline degradation enzymes by Duan et al. in the *Journal of American Chemical Society*, and development of an enzymatic pathway for nucleoside antibiotic production by McIntosh et al. in *Nature*, all use a similar approach for assessing enzyme activity via HPLC analysis and spectrophotometric assays which justifies our use of peak area and spectrophotometric assays, rather than reporting yield, as standard for this caliber of publication.

Line 199: “We confirmed that the active purified TTAs (PiTTA, CsTTA, BuTTA, KaTTA, and PbTTA) only act with L-Thr with no β -OH-nsAA formation using L-Ser or Gly (Fig. S23). Further, this result confirmed that after 20 h, ObiH, PiTTA, CsTTA, BuTTA, KaTTA, and PbTTA all approached 100% conversion of the aldehyde to the final β -OH-nsAA product.”

We also added the following comment to the caption of the associated SI figure.

S37: The percentage above the L-Thr bars indicates the analytical percent conversion observed for that reaction.

4. Generally, I find the citations given to previous work sufficient. Including the work by Wencewicz (2017) and Wilkinson (2017). I think the two citations (and maybe more) of using the ObiH for biocatalytic cascades (e.g. Andrew, *Angew. Chem. Int. Ed.* 2022, 61, e202212637) should also be included in the introduction.

We agree with the reviewer that this citation should be included in the introduction and have appended it to the following sentence in the manuscript describing the diverse array of substrates that have been tested with ObiH.

Line 68: “Since its discovery, ObiH (and a 99% similar variant, PsLTTA) has been characterized to have activity on over 30 aldehyde substrates as a purified enzyme and in resting cell biocatalysts, with notably little to no activity on aromatic aldehydes that contain strongly electron-donating functional groups²³⁻²⁸.”

Except for the few points raised above, I find the manuscript well-written and clear. The logic is very clear and precise. In my view, a revised version of the manuscript might very well be publishable in *Communication Biology*.

We are very appreciative of the positive review and would like to thank the reviewer for providing their comments.

Reviewer #2:

Nonstandard amino acids are important building blocks for the production of small molecule antibiotics and polypeptides. Therefore, harnessing the enzymatic machinery responsible for their biosynthesis is of great interest for applications in biocatalysis and, more in general, biotechnology.

In this work, Jones and coworkers set out to discover and characterize new L-Threonine Transaldolases (TTAs), which are able to perform aldol condensation reactions between L-Thr and different types of aldehydes to yield nonstandard β -OH amino acids (β -OH-nsAA). By using a multidisciplinary approach—involving computational methods such as network analysis and protein sequence/structure alignments, as well as synthetic biology tools, enzymatic assays, and analytical chemistry techniques—the authors successfully identify, express, and characterize several catalytically active TTAs distantly related to each other. Furthermore, they establish an

elegant in vivo system that can be used for the production of β -OH-nsAA (with functional handles) starting from accessible substrates such as L-Thr and carboxylic acids.

In my opinion, the research described here is of great quality and the rationale behind the experiments is solid. All the proper controls have been carried out and all experiments are presented in triplicates in a visually compelling manner. The extensive supplementary material file demonstrates that the authors approached this work thoroughly and it provides a lot of useful information to go along with the main manuscript, especially should other researchers wish to replicate some of the work. In general, I believe that the results presented in this work can lead to the development of efficient biocatalytic processes for important building blocks. Lastly, the approach used in this work can benefit other researchers who wish to discover and study other types of biocatalysts.

We would like to thank the reviewer for reviewing our manuscript and for the positive description of the work.

I only have a few points that I would like to address:

1. Lines 85-86 & Fig. 1: the authors state that they tackle specific challenges related to in vivo biosynthesis of β -OH-nsAA, including aldehyde substrate stability, and they then refer to Fig. 1c. From this figure it is not really clear to me that this particular problem is addressed (as it is instead in Fig. 5c). Maybe the figure can be adjusted accordingly.

We would like to thank the reviewer for highlighting this discrepancy. We have adjusted the cell depiction in Figure 1 to address that we have made knockouts in the strain to aid with aldehyde stability. In addition, we have added this statement to the caption of the figure.

Line 551: "... (5) The potential for TTA-catalyzed formation of beta hydroxylated non-standard amino acids during aerobic fermentation using an engineered chassis for aldehyde stabilization."

2. Lines 186-187: why is phenylacetaldehyde being used for kinetics, when one of the best performing substrates for these enzymes and assay is 2-nitrobenzaldehyde (lines 174-175, Fig. 2 and 5)? This should be explained further.

We agree with the reviewer that it is unexpected that we used phenylacetaldehyde for the kinetics measurements. We chose to use phenylacetaldehyde for a variety of reasons, the first being that it was a stable analog of the natural ObiH substrate 4-nitrophenylacetaldehyde that was previously used for L-Thr K_M measurements in the literature with ObiH. Next, we know that it is completely compatible with the TTA-ADH assay, meaning that it does not absorb at 340 nm nor does the ADH have any background activity. We believe that we would be able to calculate the L-Thr K_M using other aldehyde substrates and for future studies we expect to study how this value changes with aldehyde substrate.

Line 220: "Two differences between our assays were the substrate, phenylacetaldehyde (14) instead of 4-nitrophenylacetylaldehyde (15), and the assay format, ADH coupling rather than a discontinuous HPLC assay. We used phenylacetaldehyde for the enzyme kinetics assay because it

does not interfere with the absorbance at 340 nm, is structurally similar to the previously reported substrates for TTA screening, and can be supplied at sufficiently low concentrations that avoid detectable off-target ADH activity. While we chose phenylacetaldehyde for this investigation, we believe this analysis could be performed with many different aldehyde substrates and may yield distinct kinetic parameters.”

3. Lines 236-237: the authors state that KaTTA has a different residue in place of Y55 (ObiH numbering), but this is not what is shown in the multiple sequence alignment in Fig. S27. There it is clear that KaTTA also has a Y residue (position 96 relative to the MSA), so is this a mistake or did the authors come to this conclusion from a different analysis? This should be corrected or clarified.

We would like to thank the reviewer for highlighting this mistake. We have updated Figure 4 and Figure S28 to reflect the alignment in the Fig S27. We have also updated the text as indicated below to remove the discussion of Y55 alignment in KaTTA. We performed a multiple sequence alignment for the KaTTA cluster with ObiH and the alignment differed slightly from what we observed with the alignment for all TTAs screened (Fig S27). Upon re-analyzing the multiple sequence alignment because of this reviewer’s comment, we identified our error and have updated the main text, main figure, and SI figure accordingly.

Line 257: “However, PbTTA and KaTTA appeared to have a modified residue at E107 which is reported to perform hydrogen bonding for PLP stabilization.”

4. Fig. S27: Something appears to have gone wrong with the processing of this figure: in the second panel (positions 60-119), the residues in the very last column are not aligned and seem to overlap in some positions.

We would like to thank the reviewer for their comment and we have corrected the error in the figure.

5. Line 227: Why was NoTTA not tested with other aldehyde substrates? Is there a specific reason? Clarify if possible.

We thank the reviewer for their curiosity. We decided not to test NoTTA further because we did not observe any background activity from the L-Thr decomposition (the first step of the reaction) when testing the enzyme. Because we did not observe this activity, we suspect that what we purified is either inactive or acts with a different amino acid substrate that was not tested. We hope to investigate this enzyme further in a follow up paper on this work. We have highlighted this explanation with the following comment within the main text.

Line 213: “We did not perform this analysis for NoTTA because we did not observe L-Thr decomposition activity, and this was predictive of inactivity on the chosen substrates for both DbTTA and TmTTA. “

6. Lines 245-246: I wonder what the rationale behind this statement is. Admittedly, I am not an expert in structural biology, but I assume that the rigid structure in PbTTA would make for a stricter

substrate specificity. Which is the opposite of what the authors speculate. Maybe they can expand their thoughts further.

We are grateful to the reviewer for this comment. We agree that our statement is speculative and thus we have removed it.

Line 263: “We then aligned the published ObiH crystal structure with an AlphaFold prediction for PbTTA and observed a β -sheet within loop 1 of PbTTA (Fig. 4e). In contrast, loop 1 in ObiH is relatively unstructured and published MD simulations³⁷ of ObiH suggest loop 1 is highly flexible.”

7. Line 252: This is connected with my 3rd comment. Why do the authors say that KaTTA does not have a residue aligned to Y55? The MSA shows it quite clearly, and if you have reached this conclusions through other analyses (e.g. structural alignment) you should clearly state and show that.

We would like to thank the reviewer for once again noting this mistake and we have corrected the text by removing the text discussing the alignment at Y55.

Line 270: “As highlighted for KaTTA and PbTTA, E107 is not conserved. For E107, each cluster appeared to have a different predominant residue in that position.”

8. Fig. S28: Along the same lines as the comment above, here the authors report “no alignment” for the KaTTA cluster with respect to the residue Y55. In accordance with comments 3 and 7, this should be clarified.

We would like to thank the reviewer for once again noting this mistake and we have updated the figure accordingly. We re-analyzed the alignment for all members of the KaTTA cluster with ObiH and observed 100% alignment of Y55 throughout the cluster.

9. Lines 419-420: the use of the terms “confluence” and “confluent” is technically not correct, as it only applies to cultures that grow as a monolayer when they cover the entire adhesive surface of their culturing vessel (e.g. most types of mammalian cells). For freshly saturated bacterial cultures grown in suspension, “overnight” alone is sufficient.

We would like to thank the reviewer for this correction, we have updated the text accordingly.

Line 444: “... stocks and grown overnight in 5 mL LBL containing kanamycin (50 μ g/mL). Overnight cultures were used...”

All in all, I would like to thank the authors for the very enjoyable read, and I am very much looking forward to seeing the final version of this manuscript.

Dr. Riccardo Iacovelli

Reviewer #3:

A. Summary of key results: In this work the authors discover, characterize, and develop fermentation strategies for making various non-standard beta-hydroxylated amino acids. Authors first characterized promiscuous activity of threonine transaldolase enzyme, ObiH, and a variety of new aldehyde acceptors. Using ObiH as a starting point, they performed bioprospecting through SSNs and experimentally characterized a variety of novel TTA enzymes. Extending their results beyond biocatalysts, they showed that these enzymes can be made compatible with fermentative conditions for making beta-hydroxylated amino acids biosynthetically. Finally, they showed how all the developments from above can be used for in vivo biosynthesis of an azide-containing beta-hydroxy amino acid that has applications in genetic code expansion.

B. Originality and significance: Threonine transaldolases have recently been garnering high interest for many of the reasons highlighted in this work. In terms of significance and originality, the work here presented by the authors is the most extensive in both scope and breadth. It characterizes different 16 aldehyde acceptors across 13 different TTA enzymes (the majority of which have never been characterized before). Authors developed and validated a new coupled assay that made screening these 208 reaction conditions scalable. Importantly, quantitative kinetic data was collected for all these conditions that establish a rich dataset for future research into this class of enzymes. Where I believe this work goes beyond what others attempt is in extending results beyond biocatalysts and moving to fermentation. Authors show how fermentation using aldehyde building blocks can be made possible by using a specially engineered ADH-knockout strain. Realizing some of the shortcomings of feeding aldehydes, they cleverly show that carboxylic acids can serve as donors if coupled to overexpression of promiscuous CAR enzymes. The final result presented in this work, azido-amino acid biosynthesis, is original and interesting in its own respect for applications in genetic code expansion.

We thank the reviewer for their generous comments in appreciation of the value of this dataset, our results under fermentative conditions, the extension to carboxylic acid precursors, and the biosynthesis of an azide-containing amino acid.

C. Data and methodology: Methods used throughout the paper were appropriate and presented in good quality. Data generated was clearly reproducible and experimental conditions used were well documented. High degree of rigor was present as all data was collected with replicates, was quantitative (if standards were present), and used orthogonal analytical tools for validation (HPLC, LC/MS).

We thank the reviewer for appreciating the rigor of our study.

D. Appropriate use of statistics and treatment of data. Statistics were correctly presented but there are minor suggestions regarding what is presented and some missing detail.

- Kinetics: The authors characterize kinetics of TTA enzymes but they should provide a point of clarity in methods or text regarding aldehyde component: (1) Specify if this was done at saturation for aldehyde, (2) explicitly specify it was done at a fixed aldehyde concentration and justify, or (3) provide literature precedence for aldol condensation of TTA enzyme reaction not being rate limiting.

We would like to thank the reviewer for these suggestions for reporting our kinetic data. In response to each comment: (1) This was not performed at saturating concentrations of aldehyde due to the background ADH activity that we observed at higher (2-10 mM) concentrations. (2) We performed this assay at a constant aldehyde concentration of 1 mM because of the compatibility with the TTA-ADH coupled assay. (3) To account for the possibility that the rate of acetaldehyde co-product formation could be distinct from the rate of the desired aldol condensation, we performed traditional quenching every minute and direct analysis of beta hydroxylated non-standard amino acid formation by HPLC-UV. These HPLC measurements provided rates that were consistent with the plate reader measurement of acetaldehyde formation (Fig. S4). We have updated these details in the methods section to clarify the study to improve reproducibility.

Line 218: "...different L-Thr concentrations and non-saturating phenylacetaldehyde concentration of 1 mM (Figs. 4b, S25)."

And,

Line 222: "We used phenylacetaldehyde for the enzyme kinetics assay because it does not interfere with the absorbance at 340 nm, is structurally similar to the previously reported substrates for TTA screening and is a low enough concentration to avoid observing background ADH activity."

- Kinetics: Authors need to specify what strategy was used for calculating MM parameters in the methods section.

We would like to thank the reviewer for highlighting this, we have updated the figure caption to specify the non-linear regression that we performed to calculate the Michaelis-Menten parameters.

S39: "The line on each graph is the result of a non-linear regression for Michaelis-Menten analysis and an asymmetric confidence interval calculation using GraphPad Prism to obtain the parameters listed in Fig. 4b."

- Kinetics: SI MM plots (S36) have incorrect units for initial rates. Should be $\mu\text{M}/\text{min}$ not mM/min .

We would like to thank the reviewer for identifying this typo, we have updated it in the figure appropriately.

- Kinetics: SI MM plots (S36) authors should specify concentration of enzyme used here OR plot rate/ E_0 instead of rate.

We would like to thank the reviewer for highlighting this, we have updated the caption of S36 to include the enzyme concentrations used.

S39: "Each plot is the measurement for a different TTA with the concentration of initial enzyme listed: (a) ObiH – $0.174 \mu\text{M}$ (b) PiTTA – $0.203 \mu\text{M}$ (c) CsTTA – $0.183 \mu\text{M}$ (d) BuTTA – $0.192 \mu\text{M}$ (e) s-KaTTA – $0.178 \mu\text{M}$ (f) PbTTA – $0.200 \mu\text{M}$ ".

- Kinetics: 95% CI intervals are presented for MM parameters. I would suggest authors report standard deviation instead.

We would like to thank the reviewer for the comment. The non-linear regression analysis that we performed uses an asymmetric confidence analysis that is considered to be more accurate and so it does not produce the standard symmetric standard deviation values. Given this we have elected to keep using the confidence interval values.

E. Conclusion: Conclusion presented is supported by evidence provided in the paper.

F. Suggested improvements:

- Page 10 Line 172-179: Consider rewriting this paragraph for better narrative.

We would like to thank the reviewer for their comment. We have restructured this section to describe our initial characterization of all TTAs with 2-nitro-benzaldehyde with the both the TTA-ADH assay and HPLC analysis. We hope that this restructuring improves the narrative for these critical results in the paper.

Line 183: “Once we purified the putative TTAs, we screened them for aldol-like condensation activity.”

- Figure S27 – Clipping on the right side, please fix.

We would like to thank the reviewer for highlighting this error, we have updated the figure accordingly.

- Provide residue number of homologues when talking about alignments. Can be tabulated in the SI. I had trouble finding which residues were being referred to in alignment since the alignments shown in SI for ObiH are numbered differently.

We would like to thank the reviewer for the suggestion. We have updated Fig. S27 to include the specific residues that we identify as significant in Fig. S28 to make it easier to track within the alignment and added the following statement to the figure caption.

S44: “The residues reported to be important for catalysis and stabilization are highlighted with a black box and the labeled ObiH residue is labeled at the top of the black box.”

G. References: Appropriate credit to previous work was used.

H. Clarity and context: Overall clarity and context provided in manuscript is high. Narrative was easy to follow and results were properly contextualized with literature. Some comments that will need to be addressed are:

- Some paragraphs, noted in minor comment, could use improvements in clarity.

We would like to thank the reviewer for their kind comments on the clarity of the manuscript, with our adjustments below we hope that we have addressed the paragraphs that are unclear.

- In SI authors state: “For StTTA, the first 36 amino acids at the N-terminus were removed to improve the similarity between StTTA and ObiH.” This statement needs to be moved into the main text and justified. It is not entirely obvious why and could help explain observations. StTTA should be shown in the main text as an Nt truncation (e.g., StTTA Δ 1-36 or something similar).

We would like to thank the reviewer for highlighting this, we have moved this comment to the main text and have amended the label for StTTA throughout the text and the figures.

Example change and added text (excluding the updates to the SI figures):

Line 161: “For one enzyme from the ObiH cluster, we arbitrarily cloned a variant to contain a 36-residue truncation from the N-terminus (StTTA- Δ 36) such that its new N-terminal residue would align with the sequence of ObiH and the other candidate TTAs.”

- Figure S5 in SI: LC/MS should be better annotated. Showing raw chromatogram is not super helpful. Other chromatograms in the SI do this properly and label known peaks clearly.

We would like to thank the reviewer for their comment. We have annotated these figures similarly to how we annotate the other LC-MS/HPLC figures in the supplementary materials and updated the caption.

S19: “The top two charts represent the raw spectra, and the bottom chart is the specific mass for the peak highlighted.”

General comment:

Main text:

- Can authors comment (in a response) on these aspects of their assays and coupled assay:
 - Absorbance of pyrodoxal 5 phosphate and intermediates
 - Reduction of pyrodoxal 5 phosphate by aldehyde dehydrogenase
 - Pyrodoxal 5 phosphate serving as an acceptor for aldol condensation

We would like to thank the reviewer for their questions. We have updated the SI figure containing the absorbances of the aldehydes to include the absorbance of the pyridoxal 5'phosphate as well as other buffer components to verify that they do not interfere with the absorbance measurements. To our knowledge, there are no intermediates that would be observed with our assay format. While previous literature has used spectrophotometry to observe the intermediate quinoid formation, in our reaction buffer and volume we were unable to observe any changes in absorbance. In addition, in negative control reactions that just contain the buffer components (NADH, phosphate buffer, magnesium chloride and PLP), the addition of ADH did not produce a substantial decrease in absorbance thus we did not observe the reduction of PLP by the ADH. Lastly, to our knowledge, the literature does not show that PLP can be an acceptor for aldol condensation.

Figures:

- Figure 4: Suggest changing “initial rate” axis title name to Initial rate/ E_0 (or similar) to distinguish from initial rate calculations that are not enzyme normalized.

We would like to thank the reviewer for this suggestion and have updated the figure accordingly.

- Figure 5, 6: Suggest changing “peak area” to whatever the measurement was instead OR specify in the figure legend that it is peak area as measured by X. Peak area is vague so it is hard to know what I am looking at.

We would like to thank the reviewer for the suggestion, and we have chosen to update the captions of Figures 5 and Figure 6 as well as the caption for Fig. S7 accordingly.

Line 599 and 612: “Peak area is calculated as the area under the curve for the new peak corresponding to the product in the absorbance spectra for the appropriate wavelength from HPLC.”

Supplementary information:

- Table S1: What is the meaning of the *E. coli* strain number? If these numbers are never referenced in a meaningful way, I would suggest removing this column. Same with plasmid numbers.

We would like to thank the reviewer for their feedback. Our numbering system allows us to succinctly reference the different strains and plasmids that we create for sharing with others and for reproducibility. For example, rather than describing the entire plasmid we can reference the plasmids used for each *E. coli* strain with ease. Much like others in the field, when we deposit these plasmids on Addgene, their titles will be labeled this way, with extended descriptions of what the plasmids contain written in other sections. Additionally, because we reference some of these strains in the methods section (Line 517), these labels are important for us to maintain.

- Table S2: Since most of the primers in this table are never explicitly referenced, can the authors more clearly describe what the primers are for? Cloning? Sequencing?

We would like to thank the reviewer for their comment. We added a caption to this table to indicate whether the primers were used for cloning or sequencing. Most primers were used for cloning, but a handful were generated for sequencing.

S7: “All primers denoted FWD and REV were used for cloning whereas any primers containing SEQ were used for sequencing the associated plasmid.”

- Table S3: Protein accession number (column header) and accession numbers are wrapped. Since you are wrapping sequence, you can just make accession number column wider to avoid this.

We would like to thank the reviewer for highlighting this, we have updated the column width, so they are not wrapped.

- S11: “For StTTA, the first 36 amino acids at the N-terminus were removed to improve the similarity between StTTA and ObiH.” Authors need to justify this more as it does not make sense to want to screen natural homologues but making large arbitrary deletions.

We would like to thank the reviewer for highlighting this. In relation to the previous comment regarding StTTA, we have moved this comment to the main text and added a modifier to the StTTA label in the figures. We have also further added a comment to the appended comment in Table S3.

S12: “*For StTTA, we arbitrarily cloned a variant to contain a 36-residue truncation from the N-terminus (StTTA-Δ36) such that its new N-terminal residue would align with the sequence of ObiH and the other candidate TTAs.”

- S25: Caption title is missing a word “L-thr is required for ...”

We would like to thank the reviewer for identifying this typo, we have updated the caption accordingly.

S37: “L-Thr is required for β-OH-nsAA production...”

- S28: In the main text you said that Y55 was aligned for KaTTA, but this disagrees with what is in S27 and in this figure (S28). I think there just needs to be a bit more clarity on the alignment and its meaning.

We would like to thank the reviewer for their comment. We have updated Figure 4 and the SI figures to be consistent with the alignment in Figure S27 since this was an error.

Minor comments:

Page 2 Line 21: Should “candidate TTA gene products” be “candidate tta gene products” since it refers to the TTA gene rather than protein?

We would like to thank the reviewer for this suggestion, since we are using the abbreviation TTA to reference L-threonine transaldolases generally we do not think we need to make this change.

Page 3 Line 34: Side note: Also found in ribosomal peptide natural products like ustiloxin B in case authors wanted an example.

We would like to thank the reviewer for the suggestion and we have updated the introduction text to include ustiloxin B.

Line 35: “...naturally in many highly effective antimicrobial non-ribosomal peptides (NRPs) such as vancomycin, ribosomally synthesized and posttranslationally modified peptides such as ustiloxin B, and industrially as small molecule antibiotics and therapeutics such as amphenicols and droxidopa.”

Page 3 Line 35: Droxidopa should not be capitalized since it is not a brand name, rather a chemical name.

We would like to thank the reviewer for highlighting this. We have updated the text accordingly.

Page 3 Line 36: “Some of these molecules” ambiguous – referring to either the 3 examples stated previously or aromatic nsAAs in general. Use more precise language.

We would like to thank the reviewer for highlighting this. We have updated the text to specify the aryl β -hydroxy nsAAs.

Line 37: “Beyond their current natural and industrial uses, aryl beta-hydroxy non-standard amino acids (β -OH-nsAAs) share structural similarity with nsAAs used for genetic code expansion...”

Page 3 Line 45: I think it would be wise to be mentioning both NRP and RiPPs are potential applications, otherwise it will appear like you miss a large natural product class. Though antimicrobial peptides are mentioned, RiPPs should be explicitly stated as these are a specific subclass of natural product.

Page 3 Line 46: Mentioning how beta hydroxy groups could be installed post translationally in a RiPP would be helpful for authors here as well (since this is another strategy they can be made that is not mentioned).

We agree with both of the reviewer’s suggestions to include RiPPs as a major class of natural products where β -hydroxylated products can be found. We have updated the introduction to mention RiPPs.

Line 48: “Many naturally occurring β -OH-nsAAs are produced within NRP synthase complexes in which the active enzyme performing the beta-hydroxylation is highly specific, or post-translationally in RiPPs by hydroxylases which are poorly characterized enzymes, limiting the potential for product diversification.”

Page 4 Line 52: Suggest remove “Fortunately”

We would like to thank the reviewer for their suggestion. We have removed “Fortunately” from this line.

Page 4 Line 54: Suggest revising this statement since it is technically incorrect “catalyze the aldol condensation of L-threonine (L-Thr) with an aldehyde”. The aldol condensation occurs through a quinonoid intermediate after a retroaldol cleavage a threonyl-PLP intermediate.

We would like to thank the reviewer for the correction. We have updated the text as follows.

Line 56: “TTAs are type I pyridoxal 5'-phosphate (PLP)-dependent enzymes that catalyze the retroaldol cleavage of L-threonine (L-Thr) to form acetaldehyde and a glycyI-quinonoid intermediate that then reacts with an aldehyde acceptor to form a β -OH-nsAA.”

Page 4 Line 42-63: The phrasing “act on” when describing transaldol reactions could be made more precise by specifying saying “aldehyde acceptors” instead.

We would like to thank the reviewer for the suggestion, we have updated the text accordingly.

Line 60: “Three types of TTAs have been identified: fluorothreonine transaldolases (FTases) that act on fluoroacetaldehyde acceptors; threonine:uridine 5' aldehyde transaldolases (LipK, AmbH) that act on uridine 5' aldehyde acceptors; and L-TTAs that act on aryl aldehyde acceptors.”

Page 4 Line 66: Suggest adding commas here maybe “ObiH, and TTAs more broadly, “
We would like to thank the reviewer for the suggestion, we have updated the text accordingly.

Page 5 Line 77: The reported K_M of ObiH for L-Thr

We would like to thank the reviewer for the suggestion, we have updated the text accordingly.

Line 81: “Additionally, the reported K_M of ObiH for L-Thr...”

Page 5 Line 77: Consider revising sentence for clarity to more clearly state that it would not be compatible with fermentative conditions in *E. coli* where thr concentrations are low.

We would like to thank the reviewer for the suggestion, we have updated the text accordingly.

Line 81: “Additionally, the reported K_M of ObiH for L-Thr (40.2 ± 3.8 mM) would suggest that the reaction would not proceed well in fermentative conditions without supplementation of L-Thr since natural *E. coli* L-Thr concentrations are low (normally <200 μ M)”

Page 5 Line 84: Please do not tackle, just address.

We would like to thank the reviewer for the suggestion, we have updated the text accordingly in Line 75.

Page 6 Line 92: Suggest rewriting sentence to avoid possessive “Our best”

We would like to thank the reviewer for the suggestion, we have updated the text accordingly.

Line 97: “Remarkably, one of the best TTAs tested is annotated as a hypothetical protein and shares only 27.2% sequence identity with ObiH.”

Page 9 Line 142: “We were excited by” suggest revising since it you could sound more objective in the value of your finding “These results have important implications for”

We would like to thank the reviewer for the suggestion, we have updated this as well as other instances of this language in the text.

Line 149: “Additionally, we observed activity of ObiH on terephthalaldehyde (7) and 4-boronobenzaldehyde (13) which both contain groups that can serve as handles for bioconjugatable handles.”

Page 9 Line 148: Can the authors specify here what they think the limitations of ObiH here are? As written, it is left to the reader to interpret what is missing.

We would like to thank the reviewer for the suggestion for improving clarity. We have edited the text to specify that we are trying to identify a TTA with a higher affinity for L-Thr than ObiH.

Line 155: “We used bioprospecting as an approach to advance our understanding of the TTA enzyme class and potentially discover a TTA capable of overcoming the limitations of ObiH such as its low affinity for L-Thr.”

Page 10 Line 167: “We were excited to observe” suggest revising

We would like to thank the reviewer for the suggestion, we have updated this as well as other instances of this language in the text.

Line 176: “We observed that the tag dramatically improved the expression of 11 TTAs (**Fig. 3c**).”

Line 243: “Additionally, one of the most active TTAs, PbTTA, also maintains high activity on a diverse array of substrates, originates from a different cluster of the SSN as ObiH, and exhibits low sequence identity (30% identity).”

Page 10 Line 172: “we identified the putative TTAs with high activity” – Though mentioned later in the paragraph, authors need to specify earlier what their objective is since activity is going to be subjective towards the substrate they chose.

We would like to thank the reviewer for the suggestion, in combination with the reviewer’s comment about clarity and narrative of the paragraph we have elected to re-write this topic sentence.

Page 10 Line 177: KaTTA with and without SUMO tag statement seems out of place (as written). Leaves open the question of why you did not do all the enzymes?

We would like to thank the reviewer for the comment, we have added an additional explanation to clarify this and modified the order of the text to improve the narrative. We decided that based on the similarity of activity for KaTTA and s-KaTTA that we did not need to perform this assay for the other inactive TTAs. Additionally, we compared the AlphaFold structures for each enzyme purified and observed high structural similarity for all putative TTAs tested, including the N-terminal region so we do not expect the SUMO tag to impact each TTA differently. Finally, the

benefit of soluble expression with the addition of the SUMO tag was so significant that it did not seem worthwhile to examine the impact of the SUMO tag on all of the candidate TTAs.

Line 183: We first screened each purified enzyme with the SUMO tag fusion intact using the TTA-ADH coupled assay. Our choice to characterize SUMO tagged proteins was well justified for three reasons: (1) the predicted structures generated with AlphaFold2 suggested the N-terminal region is distal from the active site for all TTAs screened; (2) the ultimate goal was to identify better homologs for expression under fermentative conditions where tag removal would be too complex or resource intensive; (3) we tested one TTA with and without the SUMO tag to verify that the tag did not impact activity (**Fig. S22**).

Page 10 Line 172-179: I think the experiments performed in this paragraph are all fine, I think the narrative of the paragraph could use a bit of work.

We would like to thank the reviewer for the comment on our experiments, we have updated the text of this paragraph to focus on what we wanted to learn with these experiments to hopefully improve the narrative.

Page 12 Line 212-227: This paragraph reads as a medly of additional experiments. I appreciate the reporting of the negative results, and want to suggest that authors could try to (1) tie the narrative of this paragraph better together or (2) highlight why the negative results are important.

We would like to thank the reviewer for the suggestion. We have made improvements to the paragraph to try and tie the paragraphs narrative together as well as emphasize the importance of our negative results.

Page 13 Line 236: Semicolon unnecessary, start new sentence.

We would like to thank the reviewer for the suggestion, we have updated the text accordingly.

Page 13 Line 236: “Many of the active TTAs within the ObiH cluster had the same residues at these sites; however, PbTTA and KaTTA appeared to have modified residues at Y55 and E107 which are reported to perform hydrogen bonding for PLP stabilization (Fig. 4d)”. Your alignment (S27) shows KaTTA as having Y55.

We would like to thank the reviewer for highlighting this error, we have updated the text and figures accordingly.

Page 14 Line 244: If you are speculating here, please provide clarity as to what the structure-to-function link is.

We would like to thank the reviewer for their comment, since this was highlighted by reviewer 2 as well we have removed the statement from the text.

Page 14 Line 247-257: I am not sure phenomenological discussion of conserved residues is a useful topic of discussion in the absence of functional characterization.

We would like to thank the reviewer for their comment. We agree in part with this assessment, and we are interested in performing such functional characterization in a follow-up manuscript. For the purposes of this story, we still believe it would benefit readers to have their attention drawn to the differences across the clusters in case they are interested in pursuing one of the proteins in a relatively new TTA cluster. As such, we have carefully considered the reviewer comment but left some discussion in the text.

Page 15 Line 265: “Productivity” is not well defined here

We would like to thank the reviewer for identifying a point with less clarity. We have updated the sentence to specify the reaction rate.

Line 283: “...better than ObiH in growing cells because their faster reaction rate of the enzyme could enable aldehyde utilization prior to aldehyde degradation by the cell.”

Page 17 Line 315: “Expensive” is relative. I would refrain from attaching value to what a chemical supplier might charge. I would suggest

We would like to thank the reviewer for this comment and since it appears as though the comment is incomplete, we have highlighted the difference in cost between the acid and aldehyde precursors to emphasize the value of the CAR-TTA coupling without assigning specific value to the aldehyde.

Line 333: “The CAR-TTA coupling is valuable because the carboxylic acid precursor is 100-fold less costly to purchase than the aldehyde precursor and the aldehyde is likely to be toxic to cells if supplied at high concentrations.”

Page 22 Line 393: Consider changing to “Materials and chemicals” since not everything listed is a chemical.

We would like to thank the reviewer for the suggestion, we have updated the sub-header accordingly.

Page 22 Line 393: It might be good practice to include supplier headquarters location to remove ambiguity.

We would like to thank the reviewer for the suggestion, we have included the supplier headquarters for each company.

Line 415: “...MilliporeSigma (Burlington, MA, USA)... Acros (Geel, Belgium TCI America (Portland, OR, USA)... Alfa Aesar (Ward Hill, MA, USA)... Advanced Chem Block (Burlingame, CA, USA)... Cayman Chemical (Ann Arbor, MI, USA)... RICCA (Arlington, TX, USA)... Fisher Chemical (Hampton, NH, USA)... ChemCruz (Dallas, TX, USA)... Teknova (Hollister, CA, USA)... Corning (Corning, NY, USA)... GoldBio (St. Louis, MO, USA)... New England BioLabs (NEB) (Ipswich, MA, USA)... Invitrogen (Waltham, MA, USA)... Proteintech (Rosemont, IL, USA).”

Page 24 Line 443: I am fairly sure this is a typo: 0.2 nM aTc should be 0.2 μ M aTc.
Page 27 Line 486: I am fairly sure this is a typo: 0.2 nM aTc should be 0.2 μ M aTc.
Page 27 Line 491: I am fairly sure this is a typo: 0.2 nM aTc should be 0.2 μ M aTc.

We would like to thank the reviewer for noticing the typo. We have updated all instances of nM concentrations for aTc and thank the reviewer for noticing this typo.

Page 29 Line 519: I would just remove the word cartoon here, say “Depiction of...”

We would like to thank the reviewer for the suggestion. We have updated the text accordingly.

Line 544: “Depiction of potential...”

REVIEWERS' COMMENTS:

Reviewer #1 (Remarks to the Author):

I have read the revised manuscripts again. The author answers my questions and concerns. At the same time, the author also answered the questions raised by other reviewers well. Therefore, I agree with the publication of the revised manuscripts.

Reviewer #2 (Remarks to the Author):

I would like to thank the authors for considering all the points that I raised during the initial review. These were thoroughly addressed in their rebuttal letter and the corrections/improvements were seamlessly integrated into the revised version of the manuscript.

I have no further comments and I wish the authors the best of luck with the last stages of resubmission and publication.

Reviewer #3 (Remarks to the Author):

Reviewed by: Jorge Marchand

General comments regarding revisions: Authors of this work appropriately addressed major and minor comments. I hope that authors generally feel as though these changes made the manuscript stronger. Beyond the tedium of addressing my comments, I also want to highlight efforts that authors went through to address other reviewer comments as well, including depositing strains used in Addgene. Reading through other reviewer comments, clarification that seemed to have confused all of us reviewers have been addressed (alignments, structural discussion, and clarification on design choice). I share enthusiasm with other reviewers for this work and look forward to its publication.

See below for summary of my impressions on the manuscript and a remaining minor suggestion for an edit below:

A. Summary of key results: In this work the authors discover, characterize, and develop fermentation strategies for making various non-standard beta-hydroxylated amino acids. Authors first characterized promiscuous activity of threonine transaldolase enzyme, ObiH, and a variety of new aldehyde acceptors. Using ObiH as a starting point, they performed bioprospecting through SSNs and experimentally characterized a variety of novel TTA enzymes. Extending their results beyond biocatalysts, they showed that these enzymes can be made compatible with fermentative conditions for making beta-hydroxylated amino acids biosynthetically. Finally, they showed how all the developments from above can be used for in vivo biosynthesis of an azide-containing beta-hydroxy amino acid that has applications in genetic code expansion.

B. Originality and significance: Threonine transaldolases have recently been garnering high interest for many of the reasons highlighted in this work. In terms of significance and originality, the work here presented by the authors is the most extensive in both scope and breadth. It characterizes different 16 aldehyde acceptors across 13 different TTA enzymes (the majority of which have never been characterized before). Authors developed and validated a new coupled assay that made screening these 208 reaction conditions scalable. Importantly, quantitative kinetic data was collected for all these conditions that establish a rich dataset for future research into this class of enzymes. Where I believe this work goes beyond what others attempt is in extending results beyond biocatalysts and moving to fermentation. Authors show how fermentation using aldehyde building blocks can be made possible by using a specially engineered ADH-knockout strain. Realizing some of the shortcomings of

feeding aldehydes, they cleverly show that carboxylic acids can serve as donors if coupled to overexpression of promiscuous CAR enzymes. The final result presented in this work, azido-amino acid biosynthesis, is original and interesting in its own respect for applications in genetic code expansion.

C. Data and methodology: Methods used throughout the paper were appropriate and presented in good quality. Data generated was clearly reproducible and experimental conditions used were well documented. High degree of rigor was present as all data was collected with replicates, was quantitative (if standards were present), and used orthogonal analytical tools for validation (HPLC, LC/MS).

D. Appropriate use of statistics and treatment of data. Statistics were correctly presented and enough detail regarding methods were justified. In their revision of the manuscript, authors addressed all necessary points of missing detail.

E. Conclusion: Conclusion presented is supported by evidence provided in the paper.

F. Suggested improvements:

a. >>> Original comment: "Please revise: Figure S5 in SI: LC/MS should be better annotated. Showing raw chromatogram is not super helpful. Other chromatograms in the SI do this properly and label known peaks clearly." This description should be further revised to: "The bottom chart shows mass spectra (m/z) extracted at the highlighted elution time."

b. Otherwise, authors did a great job addressing all major and minor comments that I (and other reviewers had suggested). I truly believe this manuscript is much stronger after the suggested edits were put in place.

G. References: Appropriate credit to previous work was used.

H. Clarity and context: Overall clarity and context provided in manuscript is high. Narrative was easy to follow and results were properly contextualized with literature. From first round of edits, authors were able to address all areas where clarity could be improved. Among them, rewriting certain sections of the work to improve the narrative and providing more detail to figure captions to better understand methods.